# Synaptic high-frequency jumping synchronises vision to high-speed behaviour

Neveen Mansour [1,2,10], Jouni Takalo [1,2,3,10] ✉, Joni Kemppainen[1,2], Alice D. Bridges[1,2], HaDi MaBouDi [1,2], Ali Asgar Bohra[1,2], Kaja Anielska[1,2], Vera Vasas [1,2], Théo Robert [1,2], Bruce Yi Bu[4], Shashwat Shukla[4], Yiyin Zhou [5], Maike Kittelmann [6], Joke Ouwendijk[7], Judith Mantell[7], Matthew Lawson [8], Gonzalo de Polavieja [9], Elizabeth Duke[8], Aurel A. Lazar [4], Paul Verkade [7], Lars Chittka [3] & Mikko Juusola [1,2,10] ✉

During high-speed behaviour, animals must synchronise perception and action despite rapid environmental and self-generated motion. How neural systems achieve such precision remains unclear. Here we show how the housefly (*Musca domestica*) maintains visual accuracy during fast motion. Using intracellular and photomechanical recordings during saccade-like stimulation, we traced information flow from photoreceptors to large monopolar cells (LMCs). Visual neurons achieved record-high information sampling (~2500 bits·s⁻¹) and synaptic transmission (~4100 bits·s⁻¹), far exceeding previous estimates. We identify a previously unknown mechanism - *synaptic high-frequency jumping* - in which photoreceptor-LMC synapses dynamically shift transmission toward higher frequencies during saccades, extending visual bandwidth to ~1000 Hz, effectively eliminating synaptic delays, and quadrupling classical flicker-fusion limits (~230 Hz). Behavioural experiments show flies respond synchronously within ~13-20 ms, even before photoreceptor responses peak. A biophysically realistic model reveals how photomechanical-stochastic-refractory quantal sampling and synaptic transmission co-adapt with saccadic behaviour: through self-motion, flies efficiently translate image motion into temporally-precise, predictive high-speed vision.

Animals moving at high speeds must process visual information rapidly to avoid motion blur. How neural circuits achieve this - and whether self-motion helps or hinders perception - remains unclear. Answering this requires examining how innate and learned behaviours refine sensory perception to support survival. Brains, constrained by thermodynamics, genetics, and cellular biophysics, dynamically harness electrochemical, kinetic, and thermal energy to respond rapidly and accurately to internal and external signals[1]. Yet prevailing models often oversimplify neural signalling by treating neurons as static, unidirectional transmitters, neglecting the role of rapid physical movements at the ultrastructural level[2–8]. Such *morphodynamic interactions* - ultrafast, activity-dependent mechanical and structural changes in neural elements - include photoreceptor microsaccades[1,2,9–15], quantal neurotransmitter release[6,7,14–16], and

[1]School of Biosciences, University of Sheffield, Sheffield, UK. [2]Neuroscience Institute, University of Sheffield, Sheffield, UK. [3]School of Biological and Behavioural Sciences, Queen Mary University of London, London, UK. [4]Bionet Group, Department of Electrical Engineering, Columbia University, New York, NY, USA. [5]Department of Computer and Information Science, Fordham University, New York, NY, USA. [6]School of Biological and Medical Sciences, Oxford Brookes University, Oxford, UK. [7]School of Biochemistry, University of Bristol, Bristol, UK. [8]European Molecular Biology Laboratory, Hamburg Unit c/o DESY, Hamburg, Germany. [9]Champalimaud Research, Champalimaud Foundation, Lisbon, Portugal. [10]These authors contributed equally: Neveen Mansour, Jouni Takalo, Mikko Juusola. ✉e-mail: j.takalo@sheffield.ac.uk; m.juusola@sheffield.ac.uk

synaptic feedback mechanisms[17–20]. These phenomena collectively accelerate and enhance neural signalling, enabling rapid and precise perception and action[11,16].

Houseflies exemplify aerial agility[21], suggesting exceptional visual capabilities shaped by strong evolutionary pressures[22]. Historically, however, flies were presumed incapable of resolving fine visual details during rapid movements (Fig. 1a–c). Fast body and head movements (saccades) produce high angular velocities[21,23–26], which, coupled with presumably slow photoreceptor responses, were thought to blur vision[27,28]. Although compensatory head and thorax adjustments[25,26,29,30] and specialised retinal zones[31,32] partially mitigate this blur, rapid saccades were still assumed to momentarily render flies "blind"[33]. Yet this longstanding assumption contradicts flies' remarkable ability to evade threats: how could flies buzzing around your head in summer, effortlessly dodging every swat, truly have impaired vision?

To resolve this paradox, we examined how fly visual circuits respond to rapid self-motion by presenting controlled saccadic stimuli - light patterns that approximate the temporal statistics of natural saccades (Fig. 1a) - while isolating visual processing from the complexities of voluntary flight. We hypothesised that the fly visual system is not merely tolerant of saccades, but has evolved to exploit them, enabling exceptionally fast, accurate, and low-latency visual processing with minimal noise. Using a combination of intracellular recordings, photoreceptor microsaccade measurements, ultrastructural analyses, and biophysically realistic computational modelling, we explore whether visual information processing in the compound eye is not performed by static circuits constrained by optics or degraded by noise. Instead, it emerges from synergetically interacting, moving neural components that use motion itself to sample, encode, and predict visual inputs.

At the core of this process is the *morphodynamic neural superposition* architecture (Fig. 1d) of the compound eye[1,11,34], formed by R1-R6 photoreceptors and large monopolar cells (LMCs). We show how the structure-function relationships within this circuit have co-evolved with saccadic behaviour to maximise coding efficiency during rapid image motion. By integrating empirical data with simulations of photoreceptor-LMC interactions within a novel theoretical framework - morphodynamic information processing[1] - we identify and mechanistically explain a previously unknown phenomenon - *synaptic high-frequency jumping* - arising from coordinated morphodynamic sampling of rapid saccadic light changes, which enables hyperacute predictive vision.

## Results

### Multiscale experimental analysis

Experimentally, we assessed the signalling performance of housefly compound eyes - that is, how accurately, rapidly, and reliably they encode visual information - by studying both static and dynamic properties (Fig. 2a; Supplementary Notes I-III). To examine static structure, we used synchrotron X-ray imaging (i) and electron microscopy (ii) on fixed preparations to characterise the optical and ultrastructural adaptations. This included measuring the size and positioning of R1-R6 rhabdomeres across the eye and quantifying how many microvilli - photon-sampling units - each photoreceptor contains.

To assess dynamic properties, we investigated neural morphodynamics in intact living flies. Using high-speed infrared microscopy[9,11] (iii), we recorded photomechanical microsaccades within neighbouring ommatidia (iv), and applied beam-propagation modelling[11] (v) to estimate how these microsaccades move and narrow R1-R6 receptive fields locally. Finally, we used sharp microelectrodes (vi) inserted through a small corneal opening to record intracellular voltage responses of photoreceptors and LMCs to light stimuli.

### Biophysical model of morphodynamic encoding

We used experimental findings to construct a biophysically-accurate multiscale model of the R1-R6 photoreceptor-LMC network (Fig. 2c; Supplementary Notes IV). Beneath each ommatidial lens, each modelled photoreceptor sampled changes in photon flux within its anatomically realistic receptive field via photomechanical movements[2,9,11], rapidly contracting and elongating along its optical axis while shifting laterally in a complex piston-like motion. These microsaccades continuously reshaped and repositioned receptive fields in response to visual stimuli, depending on the size, eccentricity, and motion axis of each photoreceptor's light-sensing structure, the rhabdome. Each R1-R6 rhabdomere contains a distinct number of microvilli, ranging from ~41,000 to ~74,000 depending on eye location (Supplementary Notes II), each functioning as an individual photon-sampling unit[35]. The model generated macroscopic R1-R6 photoreceptor responses from quantal photon absorptions and integrated these dynamic inputs through feedforward and feedback synapses to form a morphodynamic neural superposition system that continuously adapted the information flow between photoreceptors and LMCs to maximise visual encoding.

In the real eye, thousands of these morphodynamic neural superposition units tile the visual surface, forming overcomplete, localised encoding channels (Fig. 2a-iv,v, b). Our model replicates how these units sample and process quantal visual information - i.e., changes in photon absorption rate - without requiring adjustable parameters. This framework links neural morphodynamics[1,9,11] and active sampling dynamics[36–38] to emergent coding strategies, such as network synchronisation, that enable high-speed vision and visually guided behaviour.

### Saccadic bursts amplify neural signalling

We performed electrophysiological experiments (Fig. 3a) using 'saccadic' light stimuli that mimic the rapidly changing intensity patterns experienced by photoreceptors during natural visual behaviours, such as rapid head and body rotations (Methods). These stimuli featured rapid contrast fluctuations, ranging from moderate ($c \approx 0.6$) to high ($c \approx 1.5$), across multiple temporal frequencies (20, 50, 100, 200, and 500 Hz 3 dB cut-offs; Fig. 3c). As a control and to benchmark our findings against classical studies[39–43], we also applied low-contrast ($c \approx 0.3$), bandwidth-limited Gaussian white noise (GWN) stimuli (Fig. 3c). Here, we focus on diurnal conditions, when houseflies are most active. Additional experiments using these and further stimuli - extending to even higher 3 dB cut-off frequencies and spanning a million-fold intensity range from darkness to bright daylight - are described in Supplementary Notes I (Supplementary Figs. 11–13).

GWN stimuli, traditionally used for estimating neural information capacity[39–43], are known to reduce encoding efficiency in neurons such as photoreceptors[9,44–47] and LMCs[16], which integrate quantal events (e.g., photon and neurotransmitter arrivals) through refractory sampling[9,44,45]. Each photoreceptor microvillus, a discrete photon-sampling unit, generates a quantum bump only after fully recovering from its previous phototransduction event[44,45,48]. Because GWN lacks the temporal structure of natural stimuli, it drives these sampling units into prolonged refractory states, impairing ability to track rapid photon-rate changes, and thereby reducing response amplitude and information content[9,44,45,49].

Consistent with this, photoreceptors and LMCs exhibited their strongest responses to high-contrast saccadic bursts (Fig. 3c). These stimuli comprise brief, bright events separated by short, darker intervals, which allow photoreceptors to recover from refractoriness and integrate more photon quanta efficiently[9]. In contrast, responses to low-contrast GWN stimuli were significantly smaller and decreased further as stimulus frequency increased (bottom row, right column). The 500 Hz GWN stimulus was particularly ineffective, revealing a limit

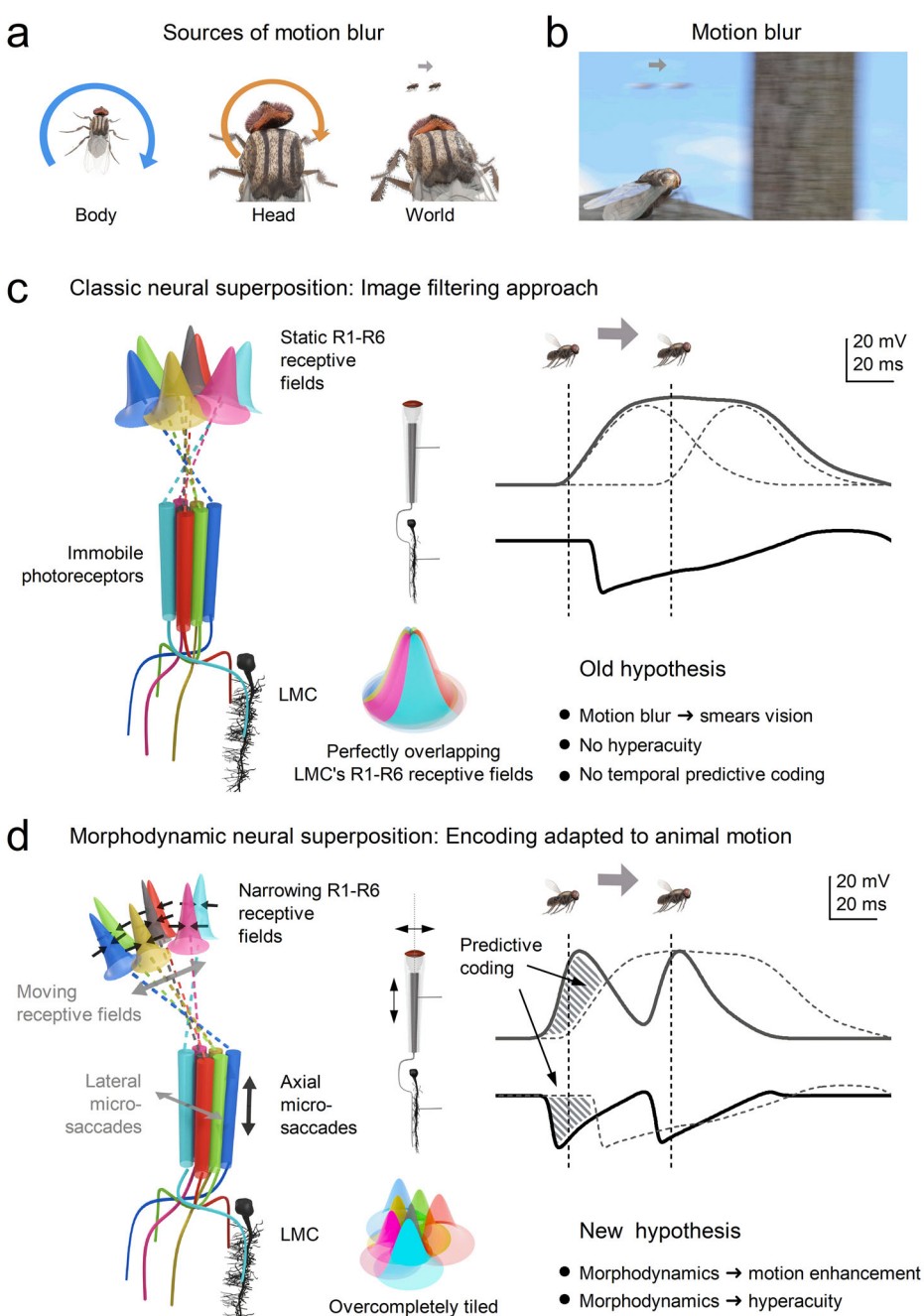

**Fig. 1 | Comparing classic static and morphodynamic models of neural superposition in fly vision. a** Sources of motion blur in natural vision arise from the animal's own rapid movements (saccadic body/head rotations) and fast motion in the external world[27,28]. These generate abrupt, burst-like retinal image shifts that challenge neural encoding. **b** Under classical assumptions, rapid image motion produces spatial and temporal smearing at the retinal level, degrading positional accuracy and limiting the resolution of fast-moving objects[27,28,33]. **c** Classic neural superposition model (image-filtering hypothesis). In the traditional view[33,96–101], R1-R6 photoreceptors within each ommatidium are immobile and functionally identical, with fixed receptive fields that overlap marginally in visual space. Large monopolar cells (LMCs) pool histaminergic[54–59] (hyperpolarising) inputs from corresponding R1-R6s in neighbouring ommatidia to form compound receptive fields representing a single spatial location[99,102] (colour-sectored dome). Encoding is modelled as a superposition of static receptive fields convolved with relatively slow temporal impulse responses[27,28]. Because receptive fields are broad and temporal integration is slow and noisy[101], responses to moving stimuli are predicted to strongly blur in space and time[33,103]. As a result, nearby moving objects cannot be resolved independently, and response peaks lag behind true object position, with no intrinsic mechanism for hyperacute localisation or predictive (phase-advanced) encoding[27,28,103]. **d** Morphodynamic neural superposition model (motion-adapted encoding hypothesis). This model incorporates measured sizes and systematic misalignments of R1-R6 rhabdomere visual axes[10,11,34], producing slightly different receptive fields that tile visual space overcompletely. Crucially, photoreceptors undergo ultrafast axial and lateral photomechanical microsaccades[9–11] that shift and narrow their receptive fields, making spatial pooling time-dependent and motion-coupled. Through stochastic-quantal-refractory sampling, this reshaping reduces noise and sharpens resolution[1,9]. Microsaccades can shift receptive fields against object motion, advancing response timing[1,9,11] (vertical dashed lines: object alignment with receptive field centres). The resulting responses enhance acuity, reduce motion blur, and generate phase-advanced temporal structure. Here, predictive coding (striped area) refers to biophysical time-locking of moving stimuli to their retinotopic representation. This mechanism differs from hierarchical error-minimisation[104–106] and efference-copy models[107–109], instead arising intrinsically from morphodynamic sampling and synaptic dynamics that minimise phase lag/preserve temporal precision during rapid motion.

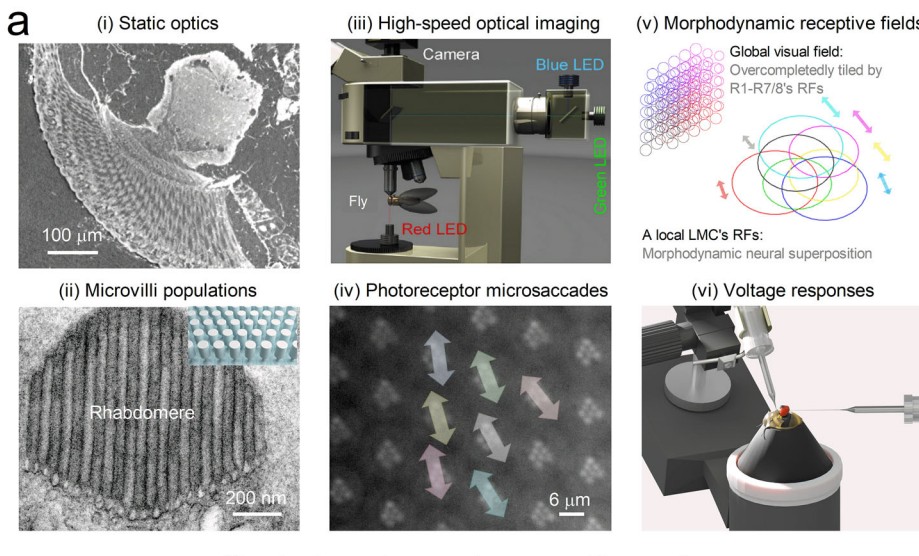

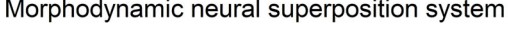

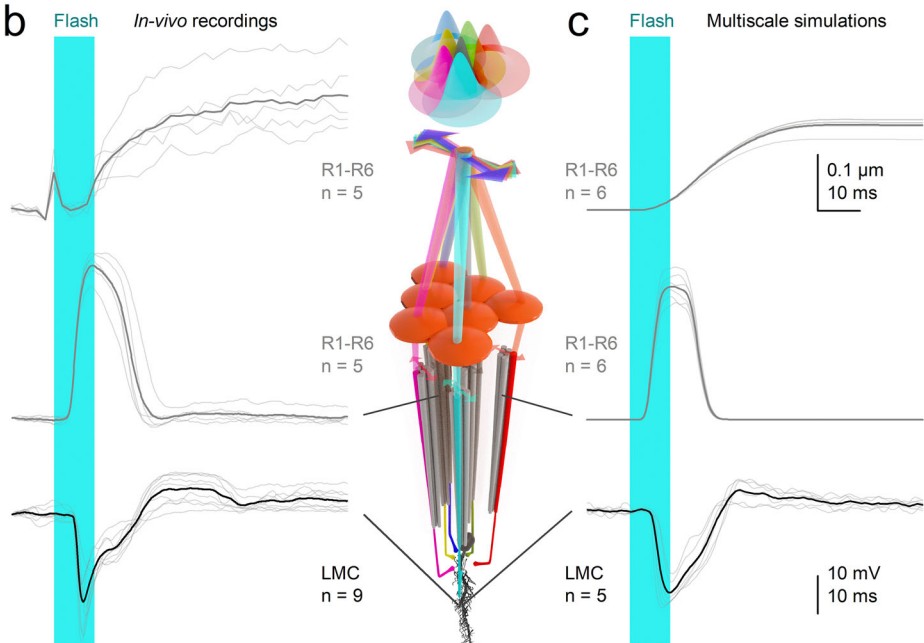

**Fig. 2 | Investigating the morphodynamic neural superposition system in the housefly compound eye. a** Structural and functional components. Key elements of morphodynamic vision were analysed using complementary approaches: (i) Synchrotron X-ray imaging of fixed eyes revealed ommatidial optical architecture (*n* = 4). (ii) Electron microscopy estimated the number of photon-sampling units (microvilli) in R1–R6 rhabdomeres (>5 flies). (iii) High-speed infrared imaging tracked in vivo photoreceptor movements; infrared light monitored motion without activation, while green/blue light evoked microsaccades. (iv) These recordings revealed photomechanical microsaccades - small, directionally diverse photoreceptor shifts within ommatidia (>10 flies). (v) Microsaccades shift photoreceptor receptive fields (RFs; beam-traced), which overlap in an overcomplete arrangement across the visual field[11,34], rather than aligning perfectly as assumed previously[97,102] (Fig. 1c,d). Each LMC integrates these dynamically moving R1–R6 inputs within a morphodynamic neural superposition system. (vi) Intracellular recordings of photoreceptor and LMC voltage responses were obtained using sharp microelectrodes. **b** Experimental responses. Responses to a 10 ms UV flash (cyan bar) show submicrometre microsaccades (top; including onset artefact), rapid photoreceptor voltage changes (middle), and downstream LMC responses (bottom). R1–R6 photoreceptors sample independently, producing variable, phase-shifted signals that converge onto a shared LMC, generating a transient, morphodynamically shaped response (thin traces, individual; thick traces, mean). **c** Model predictions. A biophysically realistic morphodynamic model, informed by (a), reproduces observed microsaccades and voltage responses. Overcompletely tiled, moving receptive fields[11,34] predict signal propagation through the photoreceptor–LMC circuit. Simulated R1–R6 responses (from one ommatidium) incorporate rhabdomere size differences (microvillar number), and simulated LMC responses show variability comparable to recordings. Responses are larger and faster, with stronger off-responses, in the anteriofrontal eye region (Supplementary Fig. 22), where increased retinal thickness and microvillar number enhance signal-to-noise ratio and accelerate responses[9,44].

in how well these neurons can track rapidly changing, randomly ordered light contrast changes.

Together, our results from individual neurons and across populations (Supplementary Figs. 2–13; Supplementary Tables 1–6) show that housefly photoreceptors and LMCs preferentially encode fast, burst-like changes in light intensity, similar to those encountered during high-speed saccadic movements[9]. These findings highlight a fundamental limitation of neural coding under unnatural steady-state conditions (e.g., prolonged exposure to bright backgrounds with GWN stimulation), which artificially elevate

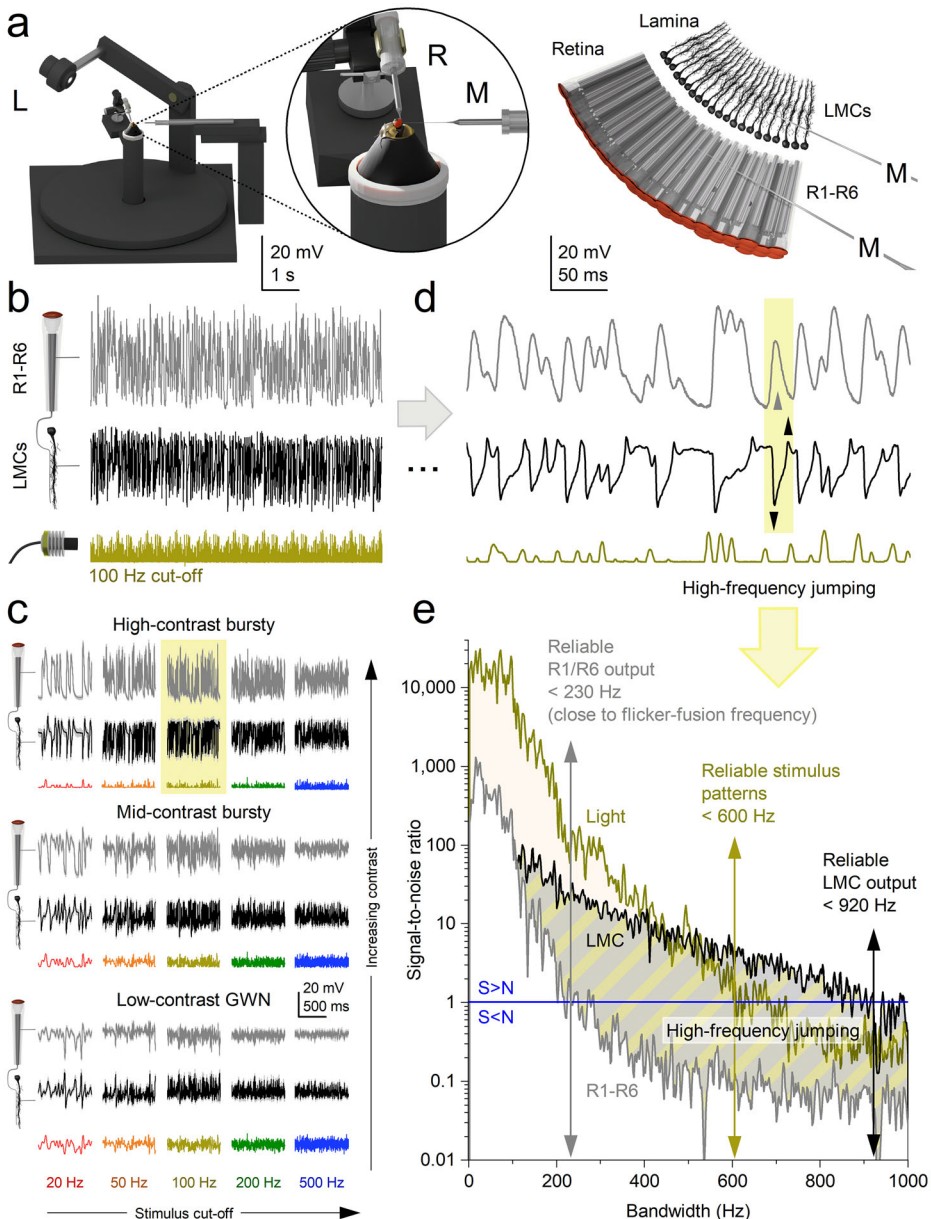

**Fig. 3 | Electrophysiological recordings:** *synaptic high-frequency jumping transfers saccadic light information from photoreceptors to higher-frequency LMC responses.* **a** Left-schematic: The Cardan arm system positions light-point stimulus (L) at the centre of the recorded neuron's receptive field, while micromanipulator-controlled measurement (M) and reference (R) microelectrodes record intracellular voltage responses. Right-schematic: R1-R6 photoreceptor and LMC recordings are conducted separately from the retina and lamina of intact, head-fixed flies (with intact eyes, red in the inset). Data from a single, stable photoreceptor and LMC with high signal-to-noise (see Supplementary Figs. 2–13 for the corresponding population data and Supplementary Tables 1–6 for the corresponding statistics). **b** Typical photoreceptor and LMC voltage responses to 30 repetitions of 2-second-long 100 Hz high-contrast bursts. **c** Photoreceptor and LMC responses to varying stimuli, from saccade-like high-contrast bursts to the maximum-size Gaussian white-noise (GWN) stimuli (of low contrast) across specific bandwidths (20, 50, 100, 200 and 500 Hz). For details, see Methods (Fig. 10). Mean

(thick traces) and individual responses (thin) to repeated stimuli (coloured). The yellow box highlights the most varying responses to 100 Hz high-contrast bursts. Recordings are from the same photoreceptor and LMC. **d** A 600 ms segment of the recordings (from **b**) illustrates how an LMC generates ultrafast, inverted biphasic on/off responses (down- and up-arrows) to the slower, more monotonically rising and falling photoreceptor responses. While the photoreceptor outputs reflect low-frequency changes in light intensity, the LMC transforms this input into high-frequency transients, effectively *transposing* the slow signal into a faster carrier band. **e** Signal-to-noise ratios of the bursty light stimulus and the resulting photoreceptor and LMC responses (from **b** and **d**) demonstrate how synaptic high-frequency jumping causes LMC responses to effectively double the base frequencies of the original light stimulus patterns. This results in LMC responses displaying an effective signalling bandwidth of approximately 920 Hz (signal-to-noise ratio >1), compared to the photoreceptor responses and the light stimulus at approximately 230 and 600 Hz, respectively.

refractoriness, reduce quantal integration efficiency, and suppress neural responses[44,46,47].

## Synaptic high-frequency jumping accelerates vision

Because rapid saccadic flight behaviours are often thought to momentarily "blind" flies through motion blur[33], we investigated

how accurately the housefly photoreceptor-LMC superposition system encodes fast-changing visual inputs. Fig. 3d illustrates, with high temporal resolution, how typical R1-R6 photoreceptors and LMCs respond to a bursty sequence of saccadic light fluctuations. Both photoreceptor and LMC responses were nearly noise-free

(Supplementary Figs. 2b, 5b), faithfully tracking repeated bursts of contrast. However, their voltage waveforms differed markedly.

Photoreceptors responded to stimulus intensity with relatively smooth, continuous signals. In contrast, LMC responses consisted of sharp, ultrafast transient signals, precisely aligned to the rising and falling phases of the photoreceptor response (Fig. 3d, yellow box with up/down arrows). These transients were temporally locked to contrast changes and effectively segmented the input signal into a string of biphasic events.

Comparing stimulus and response bandwidths revealed that LMC signals consistently shifted toward significantly higher frequencies, reliably encoding information up to ~1000 Hz (signal-to-noise ratio > 1; Fig. 3e; Supplementary Fig. 11). This *synaptic high-frequency jumping* far exceeded both the reliable stimulus bandwidth (~600 Hz) and the photoreceptor's encoding limit (~230 Hz in this example). Thus, the biphasic nature of LMC responses (Fig. 3d, below) can quadruple or more the frequency content of their photoreceptor inputs (above), efficiently and instantaneously accentuating transitions in the photoreceptor signal to enhance temporal precision, enabling the synapse to resolve events with ~0.5 ms precision (Supplementary Fig. 11; Supplementary Movie 1).

Thus, during rapid saccadic visual input, the photoreceptor-LMC circuit employs high-frequency jumping to accelerate vision - shifting neural signals into higher-frequency carrier bands where fast transients can be more effectively represented and transmitted - a strategy that mitigates motion blur and supports high-speed, predictive control of behaviour.

## High-frequency jumping maximises neural information

We next investigated how high-frequency jumping affects information transfer between R1-R6 photoreceptors (Fig. 4a) and LMCs (Fig. 4b) under diverse visual stimuli. To simulate realistic conditions encountered during rapid, cluttered flight, we used ten saccadic light patterns and five randomised Gaussian-white-noise (GWN) light patterns as controls (Fig. 3c).

Photoreceptor responses (Fig. 4c) showed that faster saccadic stimuli maximally broadened their effective signalling bandwidth (signal-to-noise ratio > 1) to ~440 Hz, with this range being limited by each photoreceptor's total number of microvilli (sampling-units)[9,44]. Variability across recordings (Fig. 3e) reflected natural differences in microvillar number, which depend on the rhabdomere length/thickness and vary across the compound eye (Supplementary Figs. 21–23). This bandwidth expansion substantially increased photoreceptor information content (Fig. 4c, left). Notably, these intracellular recordings typically exceeded the classic flicker-fusion frequency for *Musca* (~230 Hz)[50], which was originally derived from electroretinograms. Such extracellular field potential measurements underestimate local neural performance by averaging spatial and temporal signal variations, background activity, and noise across the eyes[18].

LMC responses (Fig. 4b) exhibited even stronger phasic activity under the same conditions. Synaptic high-frequency jumping redistributed photoreceptor signals into higher-frequency carrier bands at the photoreceptor-LMC synapse, greatly extending LMC bandwidth (Fig. 4c, right). This effect was strongest during high-frequency, high-contrast bursts, where photoreceptor signal-to-noise ratios reached ~2000 (Fig. 4c, left). Under these conditions, LMC bandwidth (~1000 Hz) more than doubled that of the corresponding photoreceptors (~440 Hz), whereas GWN stimuli (Fig. 4d) produced only modest increases (photoreceptors: ~210 Hz; LMCs: ~255 Hz).

The degree of LMC bandwidth extension depended on stimulus intensity, contrast, and cut-off frequency, with high-frequency jumping reaching maximal bandwidths during 50–200 Hz high-contrast bursts (~920–1000 Hz) at daylight intensities (Supplementary Figs. 11–13). For the LMC shown (Fig. 4c), all high-contrast bursty stimuli except the slowest (20 Hz) evoked high-frequency jumping,

whereas low-contrast GWN never did. These results underscore GWN's limitations in evaluating naturalistic neural coding[1,9,44,47], particularly for fast, behaviourally relevant inputs.

With improved high-frequency signal-to-noise ratios, photoreceptor information transfer rates (Fig. 4e, left) peaked at ~1200–2500 bits·s⁻¹ during 200 Hz saccadic stimulation - compared to ~600–1000 bits·s⁻¹ under GWN. Corresponding LMC rates (Fig. 4e, right) were 2-3 times higher, reaching ~2500–4100 bits·s⁻¹. These are likely the highest neural information rates reported to date and more than double those previously measured in *Calliphora* photoreceptors and LMCs under GWN[39]. Thus, the housefly's morphodynamic neural superposition system appears explicitly tuned to encode fast saccadic inputs with exceptional efficiency and minimal noise, far surpassing conventional expectations[42,51,52].

While our intracellular recordings clearly demonstrate the crucial role of synaptic high-frequency jumping in maximising information transfer during saccadic stimulation - mimicking information flow during high-speed behaviours - they cannot fully explain the underlying biophysical mechanisms. To address this, we systematically tested our multiscale model of the morphodynamic neural superposition system (Figs. 2c, 5), directly comparing its predictions with experimental intracellular recordings and performance analyses.

## Multiscale Interactions Induce High-Frequency Jumping

The morphodynamic photoreceptor-LMC neural superposition model (Fig. 5a, b) accurately replicates experimental data and elucidates the mechanistic origins of high-frequency jumping (Fig. 5c; Supplementary Figs. 27–38). Simulations show that LMCs' transient responses (Fig. 3b) - and thus high-frequency jumping - arise during the parallel tonic[16–18,53] quantal histamine release from six photoreceptors (R1-R6) into a shared LMC[54–59]. Importantly, high-frequency jumping is not the result of a single mechanism but emerges from concurrent adaptive interactions between pre- and postsynaptic processes, as illustrated by two major circular feedback loops (Fig. 5a).

The model reflects the compound eye anatomy: R1-R6 photoreceptors from neighbouring ommatidia have partially overlapping receptive fields, forming a "flower pattern"[11,34] (Fig. 5a). In the first feedback loop (top arrow circle), these receptive fields react to a spatiotemporal light stimulus (green disk) with microsaccades - tiny, directionally varied shifts driven by stochastic refractory photon sampling and photomechanical transduction in ~41,000–74,000 microvilli per photoreceptor. This morphodynamic sampling generates diverse quantum bump sequences and voltage responses across R1-R6 (Fig. 5b, top three traces), dynamically tuning each photoreceptor's receptive field to local stimulus changes.

In the second feedback loop (bottom arrow circle), differences in R1-R6 voltages modulate their tonic histamine release probabilities. Histamine binding to LMC receptors triggers Cl⁻ influx (left) and produces hyperpolarising postsynaptic responses[16,18,19,54–59]. The LMC thus integrates quantal input from six photoreceptors with partially overlapping receptive fields, boosting visual information flow to the brain[16,18,60,61] (while suppressing aliasing[9,11]). Simultaneously, LMC signals provide excitatory feedback to photoreceptors, maintaining their tonic readiness. By continuously balancing excitatory and inhibitory loads[17–19,53,62,63], this feedback ensures phasic and undelayed synaptic transmission.

To identify the components essential for high-frequency jumping, we systematically disabled or modified key mechanisms within the model. Simulations, consistent with the data processing theorem[46,64], revealed that pooling signals from all six photoreceptors is critical (Fig. 5d). A single photoreceptor-LMC synapse, even under ideal noise-free conditions, transmits no more information than the photoreceptor itself. Tonic quantal release alone - e.g. in a single R1-LMC connection -introduces background noise, reducing the LMC's signal-to-noise ratio and preventing high-frequency jumping.

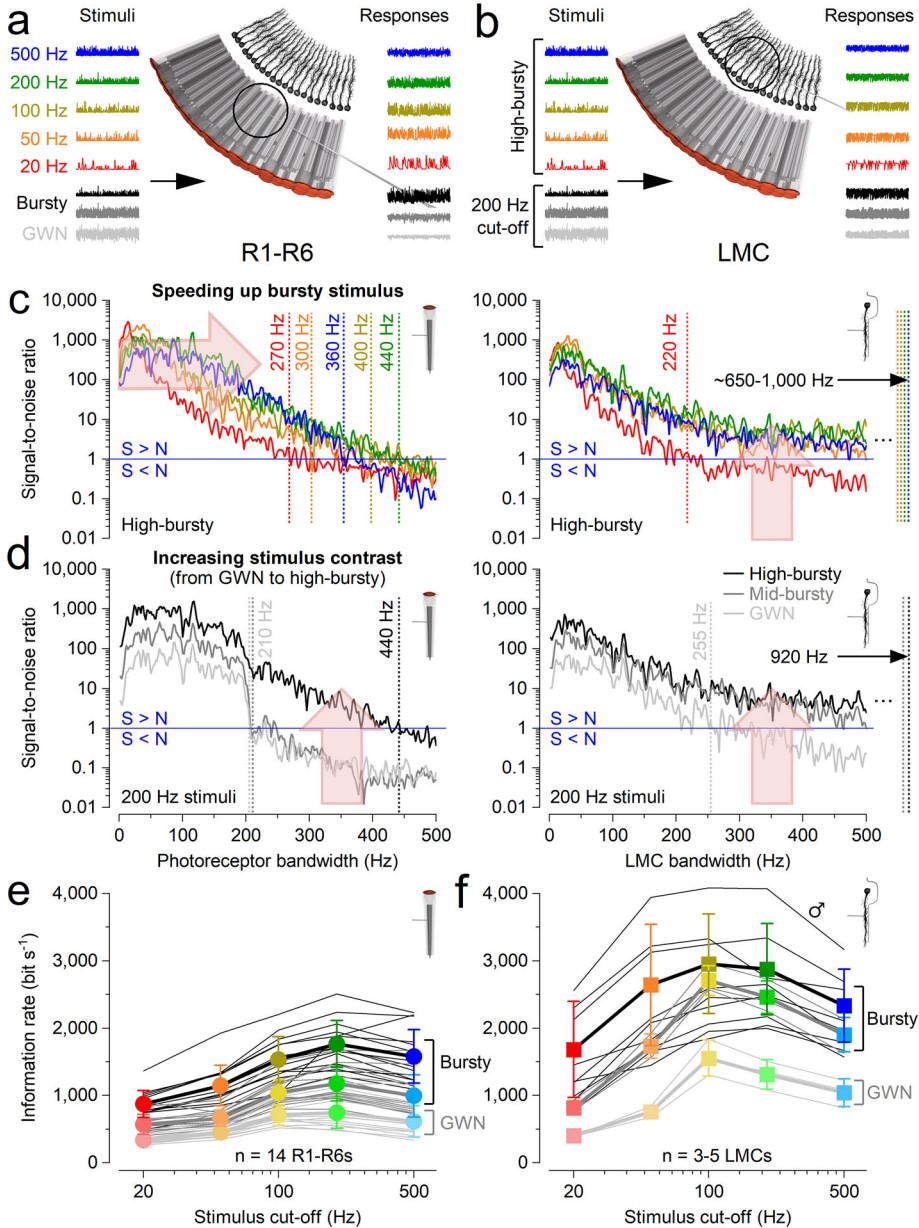

**Fig. 4 | Electrophysiological recordings: high-frequency jumping maximises information transfer to LMCs during fast, high-contrast saccadic bursts. a–d** To ensure consistent depiction of stimulus-dependent adaptation trends, we present data from a single, highly-stable R1-R6 photoreceptor and LMC recording series. **e, f** Population data: photoreceptors (*n* = 17) and LMCs (*n* = 5) in Supplementary Figs. 2–13. **a** R1-R6 signalling performance was calculated from responses to repeated 2-s light patterns across 15 different stimuli. **b** Similar LMC analyses. **c** Left: in R1-R6s, higher saccadic stimulation cut-off (right-arrow) expands effective signalling bandwidth up to ~440 (dotted lines, signal-to-noise ratio >1), increasing information content. Right: in LMCs, high-frequency jumping reallocate this enhanced information across their entire bandwidth (up-arrow), elevating the signal-to-noise ratio and achieving effective signalling up to ~1000 Hz (excluding the slowest 20 Hz bursts, red trace). **d** Left: Photoreceptors' effective bandwidth massively expands with stimulus contrast. For high-contrast saccadic bursts (c ≈ 1.29), the maximum signal-to-noise ratio reaches ~2000, and the bandwidth doubles to ~440 Hz, compared to low-contrast Gaussian white noise (GWN;

c ≈ 0.33) where the maximum signal-to-noise ratio is ~100 with an effective bandwidth of ~210 Hz. Right: LMCs' effective bandwidth also increases and broadens with stimulus contrast. High-frequency jumping is notably more effective during high-contrast bursts than with GWN. Consequently, during saccadic stimuli, LMCs' effective signalling range extends to over twice that of photoreceptors, reaching ~920 Hz (200 Hz high-contrast bursts). Whereas with GWN, the LMCs' effective signalling range is only slightly wider, ~255 Hz, compared to the photoreceptors' ~210 Hz. **e** Photoreceptors' information rates peaked for 200 Hz high-contrast "saccadic" (bursty) stimulation, with the highest estimates reaching about 2500 bits·s⁻¹. Their information rates during GWN stimulation were 2-to-3-times lower. **f** LMCs' information rates were 2-to-3-times higher than those of photoreceptors, reaching up to 4000 bits·s⁻¹ (one male fly). These estimates typically peaked for 100–200 Hz high-contrast "saccadic" bursts. The corresponding information rates during GWN stimulation were 2-to-3 times lower. **e, f** Thin line, individual cells; thick, mean ± SD. The cell-to-cell variations in information estimates likely reflect variable microelectrode recording locations and the eye's sexual dimorphism.

In contrast, pooling slightly variable conductance changes from six photoreceptors (Fig. 5b), each with a signal-to-noise ratio >1000 (Fig. 4), naturally cancels this noise[9]. Pooling increases the collective photoreceptor output six-fold, driving the synapse to progressively

"clip" the extremes of these near-noise-free bursts (Supplementary Notes IV.11). This clipping produces a square-like waveform, injecting high-frequency (>500 Hz) components that extend the LMC response bandwidth to nearly 1000 Hz and enable in vivo-like high-frequency

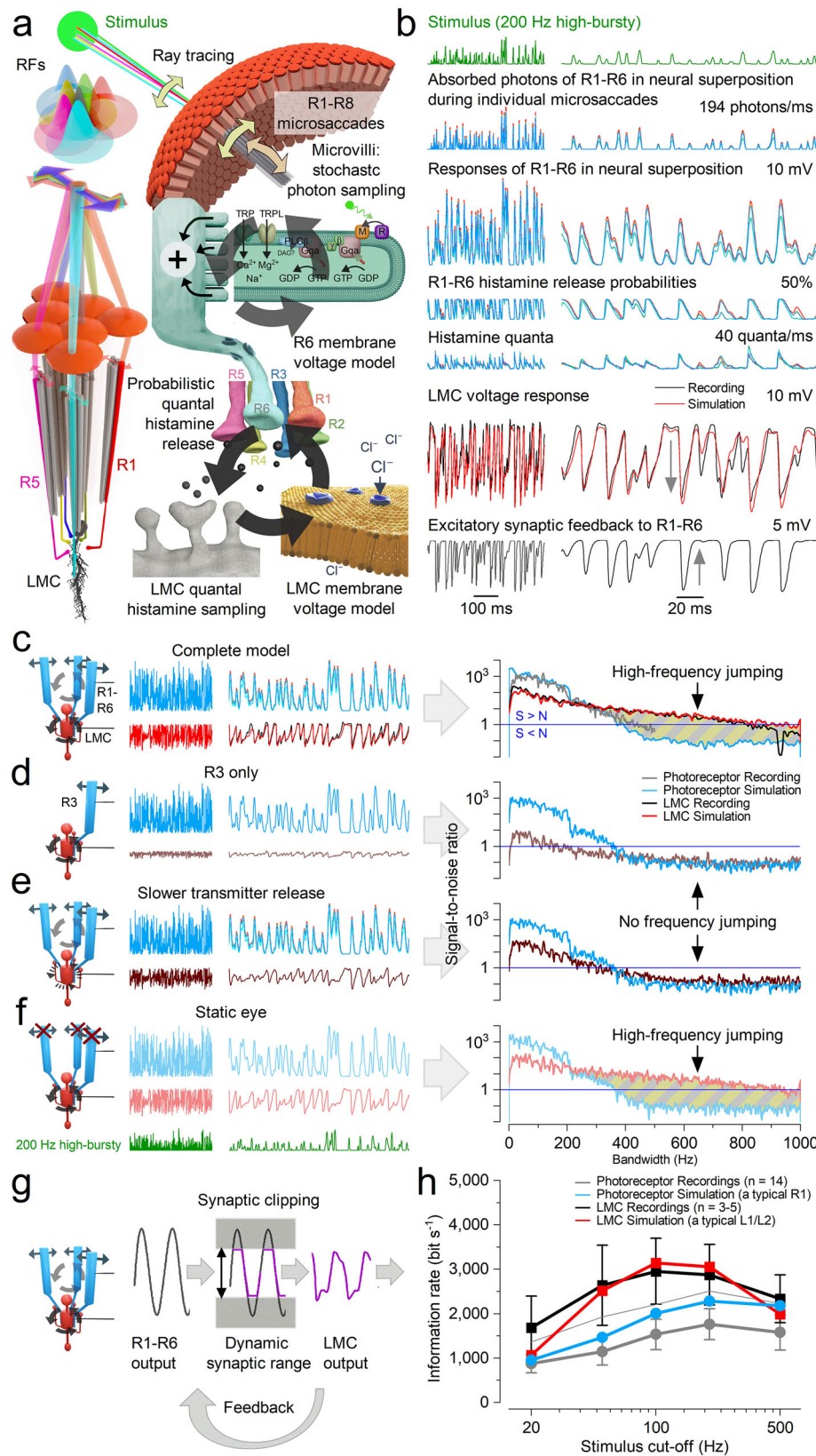

jumping (Fig. 5c; Supplementary Notes II.8; Supplementary Figs. 24, 25).

Another crucial requirement is rapid quantal transmitter release in photoreceptors (Fig. 5e). Slower release weakens modulation of histamine release probability, rendering LMC responses more low-pass and photoreceptor-like, reducing high-frequency signal-to-noise ratios and impairing high-frequency jumping. Moreover, removing excitatory feedback from lamina interneurons to photoreceptor terminals degrades both photoreceptor and LMC signal fidelity, consistent with experimental findings (Fig. 5g)[18,65]. However, synaptic high-frequency jumping still persists, albeit with altered adaptive properties (Supplementary Fig. 33d).

**Fig. 5 | Morphodynamic neural superposition model: high-frequency jumping and hyperacuity emerge from parallel photoreceptor inputs shaped by multilayered interactions. a** Model architecture. R1-R6 photoreceptors from neighbouring ommatidia sample light through a "flower-like" pattern of partially overlapping receptive fields (RFs). Feedback loop 1 (top): microsaccades dynamically shift RFs in response to local contrast, driving stochastic quantum bump sequences and adapting RF positions. Feedback loop 2 (bottom): voltage differences between R1-R6 photoreceptors trigger quantal histamine release onto postsynaptic chloride channels in LMCs, generating hyperpolarising responses[54–59]. This engages excitatory feedback from LMCs to photoreceptors, balancing synaptic loads and enabling fast, phasic transmission[18,19]. **b** Biophysical signal flow during high-speed stimulation (200 Hz). Top to bottom: photon absorption during microsaccades; photoreceptor responses; histamine release probabilities and quantal output (six traces); resulting LMC voltage; and depolarising synaptic feedback[18–20]. Simulations (blue/red) closely match in vivo recordings (black/grey). **c** Complete model reproduces high-frequency jumping. Signal-to-noise ratios (SNRs) of recordings and simulations remain high across a broad frequency range under bursty stimulation, confirming high-frequency jumping in LMCs. L1/L2 LMC subtypes receive identical histaminergic input from R1-R6[18,62,110]; their On/Off polarities emerge downstream in the medulla[60]. **d** Single input disrupts high-frequency jumping. Driving the LMC with one photoreceptor removes high-frequency jumping and reduces SNR. **e** Slower transmitter release suppresses high-frequency jumping. Reduced synaptic speed lowers temporal resolution and disrupts high-frequency structure. **f** Phase locking persists without microsaccades. A static-eye model still supports high-frequency jumping via intrinsic stochastic-quantal-refractory sampling. **g** Excitatory feedback. LMC feedback[18,19] modulates photoreceptor responses, minimising clipping within the limited synaptic output range and enhancing signal differentiation. **h** Information rates match recordings. Simulated R1–R6 and LMC responses carry similar information to recordings, peaking near 200 Hz (mean ± SD from Fig. 4e, f). Photoreceptors respond maximally to ~200 Hz bursty input, which engages refractory stochastic sampling and sharpens temporal contrast. After synaptic high-frequency jumping, LMCs respond most strongly to burst envelopes around ~100 Hz, as synaptic processing redistributes power from these lower-frequency inputs into higher-frequency components (extending toward ~1000 Hz). Thus, photoreceptor drive and LMC transmission peak at different frequency ranges.

This feedback limits the time that large photoreceptor voltage responses - evoked by high-contrast saccadic bursts - spend outside the operational range of the histaminergic output synapse. When photoreceptors transiently hyperpolarise below this range, feedback phasically depolarises them; when voltages rise above it, the feedback is reduced or switched off[18,19]. Under daylight conditions, this feedback therefore acts as an additional high-pass mechanism, broadening synaptic bandwidth and improving both photoreceptor and LMC signal fidelity.

Interestingly, the model suggests that microsaccades contribute only indirectly to high-frequency jumping. Even in simulations with static (non-moving) photoreceptors, high-frequency jumping persists if synaptic feedback is preserved, though with altered dynamics (Fig. 5f). Microsaccades introduce variability due to asymmetric, stochastic and slightly asynchronous R1-R6 movements. In a neural superposition system, this results in small timing offsets - where one photoreceptor may activate before another. This temporal variability reduces the estimated information transfer rates of both photoreceptors and LMCs by ~10%. For example, simulated LMCs transmit an average of 3481 bits·s$^{-1}$ with static photoreceptors, compared to 3142 bits·s$^{-1}$ with microsaccadic sampling (for 100 Hz high-contrast bursts: Fig. 5h).

### Photomechanical interactions accentuate predictive hyperacute responses

We therefore asked whether this modest reduction in information transmission represents a small trade-off for enhanced spatiotemporal acuity.

The visual world is not a flat surface of stationary 1/f contrast patterns, but a complex three-dimensional environment in which objects occlude one another, *shaping the visual input animals experience as they move through space* (Fig. 6a). Visual systems must therefore encode dynamic spatial structure with sufficient acuity to guide rapid behaviour under self-motion.

We hypothesised that cell-intrinsic photomechanical photoreceptor microsaccades[1,2,9–11] - arising from asymmetric and rotated rhabdomere arrangements across neighbouring ommatidia - enhance compound-eye spatiotemporal acuity at the level of LMC output by transforming motion into structured temporal cues, thereby improving the resolution of moving objects[9,11] (Supplementary Figs. 14–18,24–25). To test this, we presented *Musca* with moving dot stimuli at different speeds; complementary photoreceptor experiments using narrowing stripe stimuli are reported in Supplementary Note I.7. Here, we focus on LMC responses, which represent the output of the morphodynamic neural superposition system transmitted to the brain.

In these experiments, head-fixed *Musca* viewed two nearby dots moving at different speeds, up to saccadic velocities[9,23] (>100°·s$^{-1}$), which abruptly appeared behind a screen (Fig. 6b). We compared how well the full morphodynamic neural superposition model (Fig. 6c) and progressively reduced variants (Fig. 6d, e), culminating in the classic stationary neural superposition model (Fig. 6f), reproduced the acuity observed in recorded LMC responses.

These experiments test the extent to which peripheral morphodynamic visual processing counteracts motion blur and exhibits predictive coding (Fig. 1). The full morphodynamic neural superposition model is identical to that used in Fig. 5 - fixed and without free parameters - with only the test stimuli differing.

Across all tested conditions, intracellular recordings and the full morphodynamic neural superposition model exhibited ultra-brief response waveforms, minimal delays, and high spatiotemporal resolvability, outperforming all reduced variants. This performance is consistent with our hypothesis (Fig. 1) of morphodynamic, stochastic, refractory, quantal sampling and synaptic high-frequency jumping in the photoreceptor-LMC circuitry jointly enabling predictive coding and motion-blur minimisation.

These results demonstrate that during fast manoeuvres in cluttered natural environments with occluding objects, *Musca* can achieve hyperacute visual discrimination under conditions that would otherwise be severely degraded by motion blur. Mathematically, this slit effect - where an occluding object transiently narrows a photoreceptor's receptive field and sharpens spatial resolution -resembles synaptic clipping (Fig. 5g), a temporal mechanism that boosts high-frequency signals (Supplementary Fig. 25).

At slower velocities (42°·s$^{-1}$), information sampling and processing by the morphodynamic neural superposition system enhance the compound eye's spatial acuity[9,11] from the anatomical limit of ~2.9° (the average interommatidial angle; Supplementary Table 10) to below 0.7°, representing a more than fourfold improvement (Fig. 6). Even at saccadic speeds (168°·s$^{-1}$), *Musca* LMCs retain hyperacute resolution, resolving details as fine as 1.4° (green triangles). The tested angular velocities fall within the natural range of housefly flight[21], where straight flight segments typically involve yaw velocities up to ~150° s$^{-1}$ and saccades reach 150-2,000° s$^{-1}$. At typical forward speeds (~70 cm s$^{-1}$), the corresponding spatial separations and viewing distances are consistent with biologically plausible object detection during flight.

Together with independent photoreceptor experiments (Supplementary Figs. 15–17) showing that *Musca* photoreceptors can resolve moving edges separated by 0.9°, the conclusion is striking: the morphodynamic neural superposition system resolves moving visual

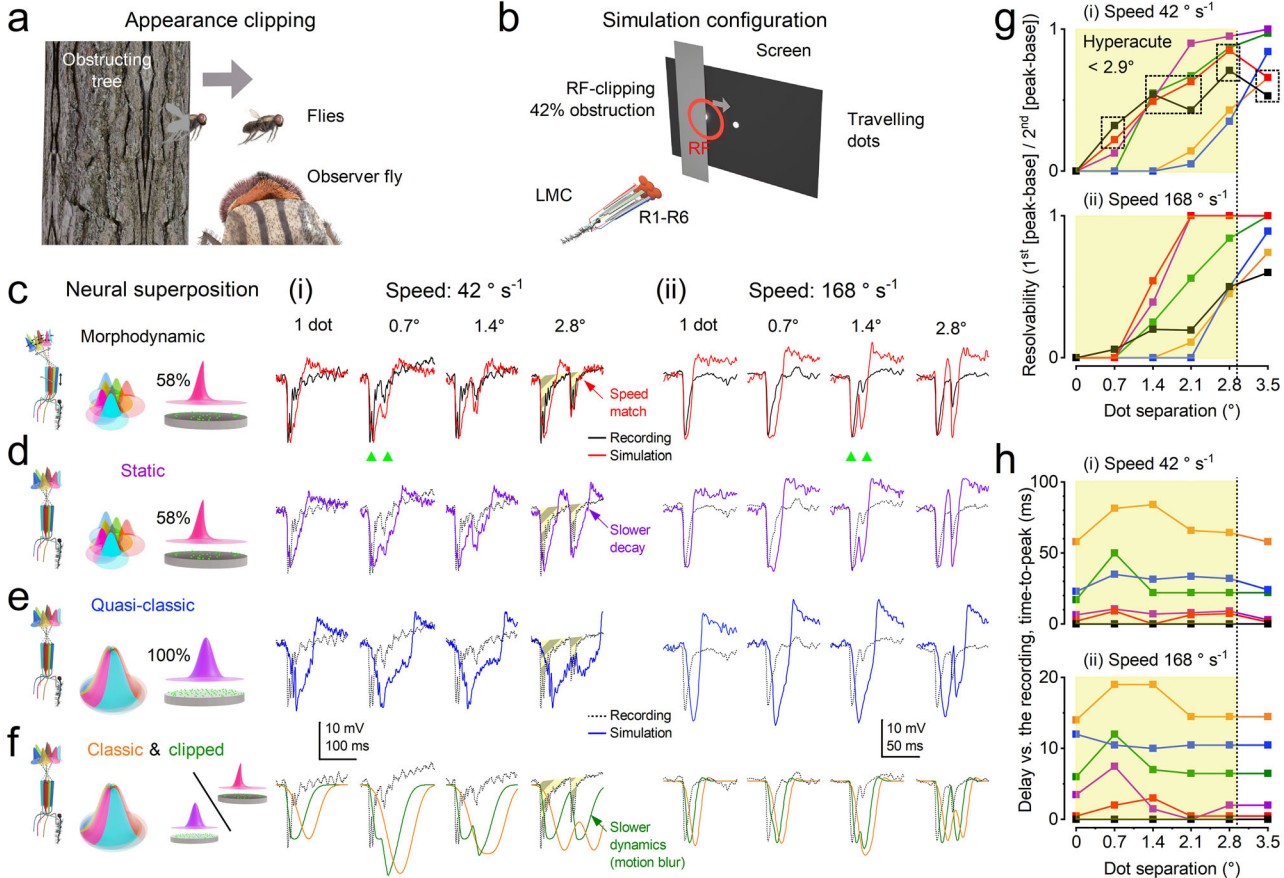

**Fig. 6 | Intracellular recordings and simulations: predicted and recorded LMC responses to occlusion-revealed moving objects under different neural superposition models.** We compared simulated LMC outputs under progressively reduced neural superposition models with recordings. In both, the LMC receptive field (RF) centre was positioned at the edge of a dark occluding plate covering 42% of the RF (centre 0.8° from the edge). One or two dots, mimicking sunlit flying insects, emerged from behind the occluder, reproducing natural conditions in which motion becomes visible after occlusion. **a** Behavioural concept. **b** Experimental/simulation-configuration: laterally moving dots emerged from behind the occluder, producing RF clipping at emergence. LMCs integrate inputs from R1-R6 photoreceptors. **c-f** Recordings/simulated LMC responses under progressively reduced models: **c** Full morphodynamic neural superposition (42% clipped RFs; red) closely matches recordings, with rapid, well-timed phasic responses. **d** Static neural superposition without microsaccades (purple), retaining overcomplete RFs, synaptic feedback, and stochastic-refractory-quantal sampling, produces slower decay and reduced spatiotemporal resolution. **e** Quasi-classical model without microsaccades (blue), with perfectly aligned RFs of identical size but retaining stochastic sampling and feedback, yields slower responses than

**c/d** However, these mechanisms still accentuate transients[9,11,17–19,44,48], reducing lag relative to the classic filter-model. **f** Classic stationary filter-model (orange/olive; full and 42% clipped RFs)[27,33,97,99,101] produces the slowest responses and poorest resolution, with little or no hyperacuity. Responses are shown for dot velocities of 42° s⁻¹ (i) and 168° s⁻¹ (ii, saccadic), with one or two dots separated by 0.7°, 1.4°, and 2.1°. Coloured traces indicate simulations; black (or dotted) traces show recordings. **g** Resolvability (first-to-second peak ratio) versus dot-separation for both speeds; dotted boxes highlight the close match between recordings (black) and the full model (red). **h** Peak-to-peak delay between simulated and recorded responses; near-zero values indicate accurate temporal prediction. The eye's average interommatidial angle (-2.9°; Supplementary Table 10) defines the static resolution limit. The morphodynamic model resolves smaller separations, even at saccadic speeds[9,23], demonstrating preserved hyperacuity during occlusion. Only this model reproduces the ultra-brief, precisely timed LMC responses observed experimentally, indicating that photoreceptor microsaccades are essential for motion-blur reduction and predictive coding. See Supplementary Fig. 18 for dot disappearance.

details at or below the diffraction limit of the average *Musca* ommatidial lens ($\theta = 1.1°$; Supplementary Notes II.8).

In summary, the second (synaptic) feedback loop in the morphodynamic neural superposition model plays the primary role in enabling temporal high-frequency jumping by coordinating phasic LMC responses to pooled inputs (Supplementary Movie 1). The first (photoreceptor) feedback loop, by contrast, primarily enhances spatial acuity, generating asymmetric "flower pattern" receptive field motion (Supplementary Movie 2) that sharpens object resolution, especially for moving objects (Fig. 6; Supplementary Figs. 15–18), and combats spatial aliasing[9,11].

### Morphodynamics matching visual lifestyles

In houseflies, as in *Drosophila*[9] and honeybees[10], photoreceptors contract photomechanically in response to changes in light intensity.

However, in *Musca*, these movements - driven by refractory photon sampling reactions[1,2,9–11,35] - occur much more rapidly, reducing saturation and more effectively maximising phasic information[9] (Fig. 7a–c). This faster refractory quantal sampling improves the signal-to-noise ratio at higher stimulus frequencies (Fig. 7e). For example, *Musca* R1-R6 photoreceptors integrate voltage signals three to four times faster than those in *Drosophila*, as reflected in their effective signalling bandwidths: -308 Hz versus -72 Hz for the same 20 Hz bursty stimulus.

Consequently, a typical R1-R6 photoreceptor in a fast-flying housefly samples approximately three times more information from the same stimulus than its slow-flying *Drosophila* counterpart, which has roughly half as many microvilli (-30,000/photoreceptor) and exhibits slower refractory and quantum bump dynamics[9,40,41,44,48]. During high-contrast saccadic bursts, *Musca* photoreceptors reach

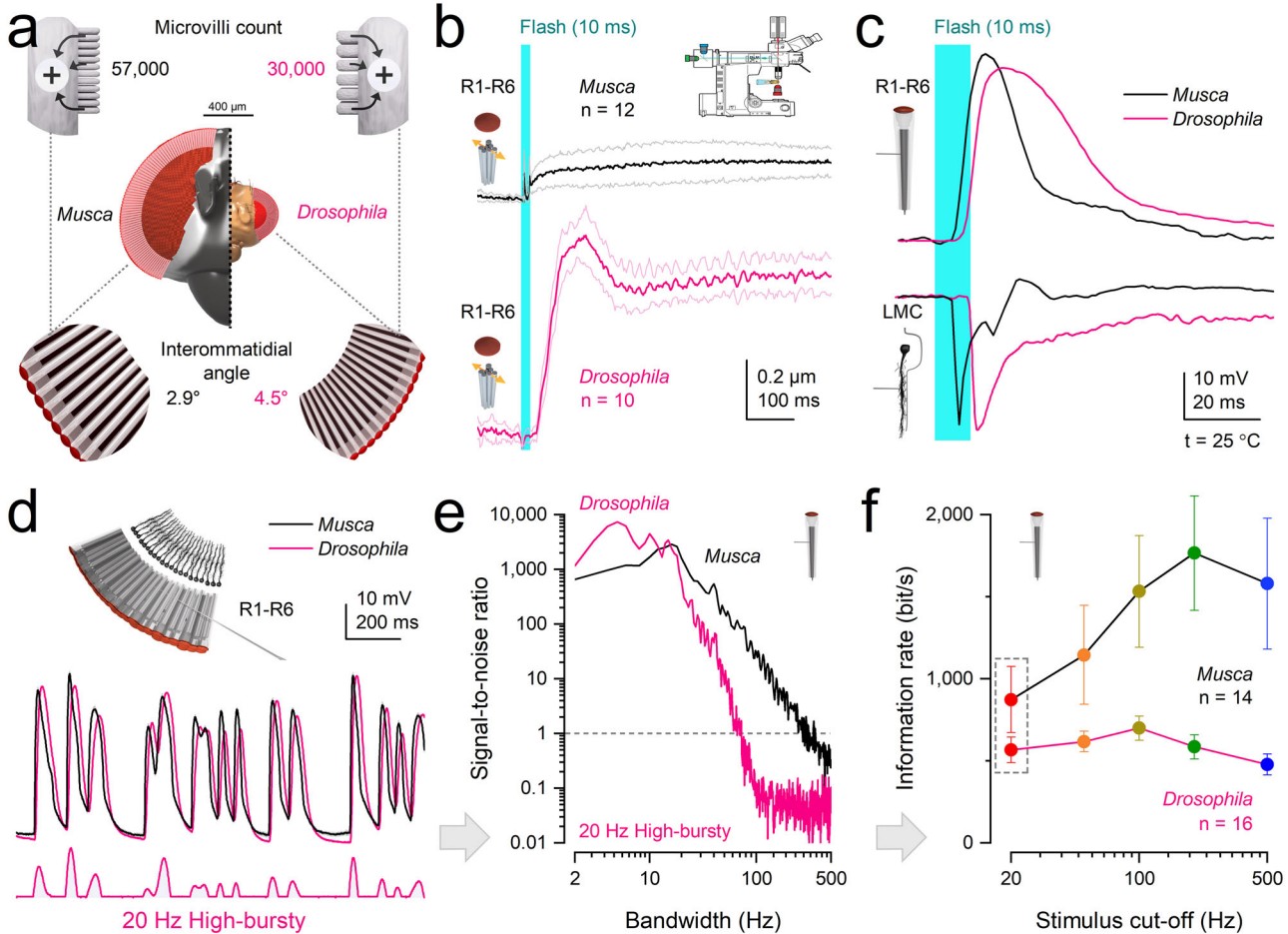

**Fig. 7 | Electrophysiological recordings: morphodynamic sampling and synaptic transmission are adapted to species-specific visual behaviours.**
**a** Anatomical specialisations. *Musca* photoreceptors contain ~54,000 microvilli · nearly twice as many as *Drosophila* (~30,000) · providing greater capacity for light sampling. The average interommatidial angle is also smaller in *Musca* (2.9° vs. 4.5°), enhancing spatial acuity. **b** Photomechanical microsaccades. *Musca* R1-R6 photoreceptors exhibit smaller, faster microsaccades than those in *Drosophila*, enabling higher temporal resolution. **c** Ultrafast neural transmission. In both species, LMC responses are biphasic and peak earlier than photoreceptor responses, with no detectable synaptic delay (Supplementary Fig. 1). In *Musca*, R1-R6 responses to a 10-ms flash reach their peak within 10–16 ms, aligning with the onset of visually guided behaviours (Fig. 8). Example recordings; population statistics in Supplementary

Fig. 1. **d** Temporal structure of saccadic responses. R1-R6 voltage responses to bursty (20 Hz) saccadic contrast changes show clear, phasic dynamics in both species, but *Musca* responses lead those of *Drosophila* in phase. **e** Signal-to-noise ratio (SNR). *Musca* photoreceptors maintain reliable signalling (SNR > 1) across a broader frequency range than *Drosophila*, supporting high-speed visual processing. **f** Information transfer. *Musca* R1-R6 photoreceptors transmit up to three times more information than those in *Drosophila*, peaking at 200 Hz for high-contrast bursts · twice the optimal frequency observed in *Drosophila* (100 Hz). Dotted box highlights the information rates for the data in (**d** and **e**). Mean ± SD. **d**–**f** Example recordings; population statistics in Supplementary Figs. 2, 3. Corresponding *Drosophila* data and analyses from[9].

maximal information rates of ~2510 bits·s$^{-1}$ (Fig. 4e), compared to ~850 bits·s$^{-1}$ in *Drosophila*[9] (Fig. 7f).

When six neural superposition photoreceptor outputs, each modulated by its own microsaccades, are combined through synaptic feedback, visual information is shifted into biphasic, aliasing-free, phase-locked LMC responses. These transiently amplify even the smallest changes in environmental contrast with minimal delay.

In both species, this rapid, bidirectional information flow eliminates classical synaptic delay: photoreceptor and LMC responses begin rising simultaneously (*Musca*-3.5 ms; *Drosophila*-6.5 ms after stimulus onset) (Fig. 7c). Yet LMC responses peak significantly earlier than their corresponding photoreceptor inputs · by ~4 ms in *Musca* and ~13 ms in *Drosophila* · consistent with predictive coding mechanisms operating at the synapse (Supplementary Fig. 1a, b).

In signal-processing terms, the photoreceptor-LMC synapse minimises phase lag relative to incoming motion trajectories, thereby improving forward encoding of dynamic input. This prediction is computational · phase-advanced and lag-minimised · rather than cognitive or expectation-based. *Musca*'s faster LMC responses (Fig. 7a–c) align with its quicker behavioural reactions and the need to time-lock visual processing to rapid motor outputs.

These findings suggest that evolution has tuned visual information processing to meet each species' behavioural demands · adjusting microvillus numbers, refractory periods, quantum bump dynamics, photomechanical responses, membrane conductances, and synaptic connectivity · while also constraining metabolic cost[9,44,45] (Supplementary Note IV). As a result, housefly photoreceptors and LMCs jointly encode contrast changes at roughly twice the speed of those in *Drosophila*[9] (Fig. 7a–c).

## Predictive coding enables ultrafast behaviours
Next, we investigated whether ultrafast morphodynamic processing, combined with high-frequency-jumping-induced acceleration of neural signalling, is also reflected in the speed of *Musca*'s visually triggered behaviours (Fig. 8). To test this, we employed two behavioural paradigms that yielded similar results. First, binocular light

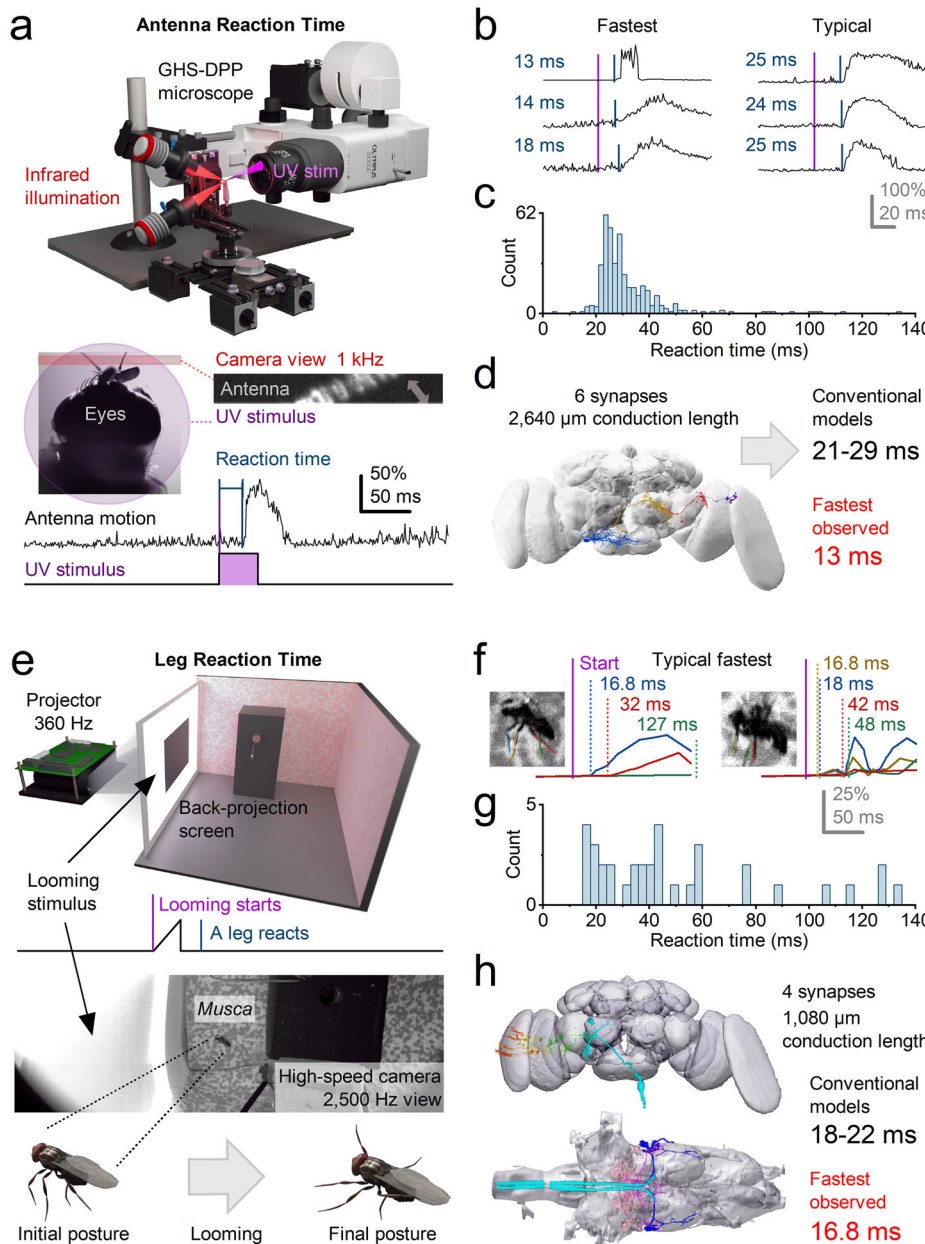

**Fig. 8 | Voluntary behavioural responses in flies outpace classical conduction-delay predictions, even along the shortest neural pathways - revealing accelerated sensorimotor processing. a** High-speed infrared videography (1 kHz) tracked *Musca* antennae responses to a UV flash. Antennal motion was recorded relative to stimulus onset to measure reaction times. **b** Cross-correlation of antennal motion revealed the shortest reaction latencies occurring within 13–18 ms (left), while more typical responses peaked around 24-25 ms (right), indicating variability across trials. **c** The broad reaction times distribution suggests that antennal responses are non-reflexive (voluntarily modulated). Some flies responded consistently, others sporadically, and some not at all. Supplementary Fig. 41 presents the statistical analysis. **d** A candidate antennal response circuit comprising 5 synapses reconstructed using the *Drosophila* connectome and scaled to *Musca* using X-ray microtomography. Standard estimates predict a minimal unidirectional pathway latency of ~21–29 ms. This is 8-16 ms (up to 2x) slower than the fastest observed 13 ms responses, implying the involvement of in vivo acceleration mechanisms (e.g., synaptic high-frequency jumping). Data: FlyWire-dataset[111] snapshot 783 accessed through Fruit-Fly-Brain-Observatory (FFBO)[112,113].

Availability: CC BY-NC 4.0-license. **e** In a separate setup, tethered flies were presented with high-speed looming stimuli via a 360 Hz projector and back-projection screen, while leg-lift responses were recorded using a 2.5 kHz high-speed camera (Supplementary Fig. 39). **f** Reaction times to looming stimuli varied substantially across individuals and trials. The fastest responses were detected after 16.8–18 ms, with others occurring at 32, 42, or even >100 ms post-stimulus, suggesting a mixture of voluntary and non-responses. **g** Distribution of leg reaction times reveals a multimodal pattern, again consistent with voluntary control. Supplementary Fig. 40 presents the statistical analysis. **h** The leg-lift pathway includes at least 4 synapses and spans ~1.1 mm in conduction length to giant fibres (GFs). Using standard assumptions (Methods), the fastest expected motor response would take ~18–22 ms. Yet observed leg responses at 16.8 ms are 1.2-5.2 ms faster than predicted, reinforcing the idea that classical serial conduction models underestimate the true speed of visual processing in active fly behaviour. Data: FlyWire-dataset[111] snapshot 783 and MANC-dataset[114] version 1.0, accessed in FFBO[112,113]. Availability: FlyWire-dataset CC BY-NC 4.0-license; MANC-dataset CC BY 4.0-license.

flashes elicited rapid antennal movements (Fig. 8a–d), potentially allowing the fly to gather additional olfactory, auditory, or thermal information to reduce stimulus-related uncertainty. Second, when startled by a rapidly looming dark object, a tethered *Musca* exhibited a synchronised, rapid lift of all six legs (Fig. 8e–h; Supplementary Movie 3).

Strikingly, the shortest response latencies ranged from 13 to 16.8 ms. These exceptionally brief delays are remarkable because the behaviours appear voluntary and decision-based, conditions typically associated with greater response variability and a higher likelihood of no response[1,66]. In both cases, neural signals must first be generated in the photoreceptors and then transmitted through brain circuits (Supplementary Movie 4). Even the most direct "reflex-like" pathway - bypassing central decision-making - would involve at least five synapses (Supplementary Notes VI) before reaching the muscles. Yet both responses were highly variable in timing and often absent, distinguishing them from classic, involuntary reflexes[67].

To estimate the minimal possible reaction time, we extrapolated from the complete *Drosophila* brain connectome, adjusting for *Musca*'s brain being approximately three times larger (Supplementary Notes VI). Using standard values for synaptic transmission, neural charging, and conduction delays within a unidirectional signalling model (Fig. 8h; Methods), we predicted a minimum reaction time of 18–29 ms for reflex-like pathways, containing four to six synapses. These predicted delays are 5–16 ms (38–123%) longer than the shortest observed voluntary response latencies.

This discrepancy suggests that *Musca*'s neural processing is not strictly feedforward, but operates within recurrent feedforward–feedback loops across brain circuits[17,18,66]. The reduction in phase lag arises locally from synaptic and morphodynamic mechanisms. However, within this broader predictive framework, the fly's internal state - including behavioural context and arousal - can modulate sensory gain and temporal coordination[1,68–70]. We propose that this global recurrent architecture supports the binding of object features across space and time, enabling predictive, phasic information flow powered by quantal, refractory, and morphodynamic mechanisms - mirroring those observed at the photoreceptor-LMC synapse (Fig. 5).

## Discussion

Integrating multiscale experimental and modelling approaches, we uncover *synaptic high-frequency jumping* and explain how it emerges. This previously undescribed mechanism enables peripheral visual neurons to shift information into higher carrier frequencies in response to high-speed saccadic input, thereby minimising communication delays and increasing the coding speed of reliable vision. Remarkably, housefly LMCs can transmit information at rates exceeding 4000 bits·s⁻¹ and operate at bandwidths approaching 1 kHz - far beyond classical flicker-fusion limits[50]. Using ultrahigh-speed videography, we further show that houseflies initiate voluntary, stimulus-triggered behaviours at a time when photoreceptor responses are only just reaching their peak.

These findings challenge long-standing models of sequential neural transmission and reveal how vision dynamically adapts to behavioural demands. Rather than passively processing visual information, houseflies actively shape their sensory input through high-speed flight behaviours, generating the spatiotemporal structures that drive high-frequency jumping, predictive coding, hyperacute perception and rapid neural synchronisation with environmental dynamics.

By highlighting the critical roles of saccadic visual behaviours, morphodynamic mechanisms, and bidirectional synaptic interactions in enabling fast, parallel, low-latency information sampling and processing, our results have broad implications for understanding efficient encoding and predictive coding. In particular, synaptic high-frequency jumping suggests a neurophysiological solution to the binding problem - that is, how information encoded across distinct brain circuits is synchronised to produce unified perception, decision, and action - within the physical constraints of neural computation.

*How does active, high-speed self-motion enhance visual acuity and information throughput?* Houseflies maintain superior visual performance during rapid saccadic turns generated by flight manoeuvres. These self-induced movements do not impair vision; rather, they enhance it. Morphodynamic neural superposition enables photoreceptors and downstream circuits to extract temporally structured, behaviourally relevant features with minimal delay, enabling simultaneous efficient processing of high-speed visual information and hyperacute perception. The resulting phasic signals are rapidly amplified and undergo synaptic high-frequency jumping, emerging as transient, biphasic LMC responses that broaden bandwidth and shorten latency. These signals are further accelerated by the brain's bidirectional information flow, where tonic feedforward (inhibitory) and feedback (excitatory) interactions help balance synaptic load[1,17–19,63]. As a result, they become synchronised with internal motor states, generating a predictive, time-locked encoding of environmental dynamics - facilitating high-speed decision-making and stable visual perception, even under variable lighting conditions.

Information throughput increases with the number of samples, assuming constant conditions[1,45,46,64]. Photoreceptors continuously adapt to fluctuating light intensities - reflecting logarithmic changes in environmental photon rates - through *refractory quantal sampling*, which dynamically adjusts the quantum efficiency of microvilli[1,9,44,48]. This mechanism enables signal-to-noise ratios of ~1000–8000 (Fig. 7e) in response to fast saccadic contrast changes under bright daylight conditions. Approximately 54,000 microvilli per cell engage in stochastic sampling, effectively absorbing ~$10^5$–$10^6$ photons per second. In bright illumination (>$10^7$ photons/s), microvillar refractoriness prevents most absorbed photons from eliciting a quantal response[1,9,44,45,48]. Under these conditions, photoreceptor microsaccades - together with the fly's saccadic flight behaviours - selectively enhance the sampling of contrast changes[1,9,11]. As a result, photoreceptors produce accurate anti-aliased estimates of the dynamic visual scene without saturating their biophysically limited amplitude or frequency ranges, while averaging out residual noise[1,9,44,48]. Their histaminergic synapses can thus respond reliably and efficiently to even the slightest contrast changes.

Each LMC integrates slightly variable inputs from six photoreceptors, or seven in the male's frontal-dorsal "love spot" region[50], via overlapping, photomechanically moving receptive fields (Fig. 9). This morphodynamic pooling of parallel photoreceptor outputs enhances spatiotemporal resolution, enabling hyperacute pattern detection and object tracking well below the ~2.9° limit imposed by the average interommatidial angle, which defines the static pixel resolution of the compound eye. Remarkably, our recordings and simulations show that the morphodynamic neural superposition system, evident in LMC voltage responses, can resolve moving objects separated by just 0.7° - narrower than the ommatidial lens's airy-disk angle (1.1°), the theoretical diffraction limit (Fig. 6, Supplementary Fig. 24) - a performance once thought impossible[33].

Notably, high-frequency jumping is absent in recordings using Gaussian white noise (GWN; Fig. 4d), which increases refractoriness and desensitises phototransduction[9,44]. It is also missing from responses to linearly presented naturalistic image time series, dominated by slow frequencies ($1/f$ statistics)[9,18,46,71]. Our results therefore suggest that animals' *active vision* - combining high-speed saccadic movements with brief fixation pauses[9], including photoreceptor microsaccades[11,12,27], intraocular muscle contractions[72], and eye, head, and body rotations[36] - actively drives high-frequency jumping. This, in turn, enhances the phase congruency of visual features (such as edges, occluding objects, changing textures, and outlines), making them stand out instantaneously in the scene.

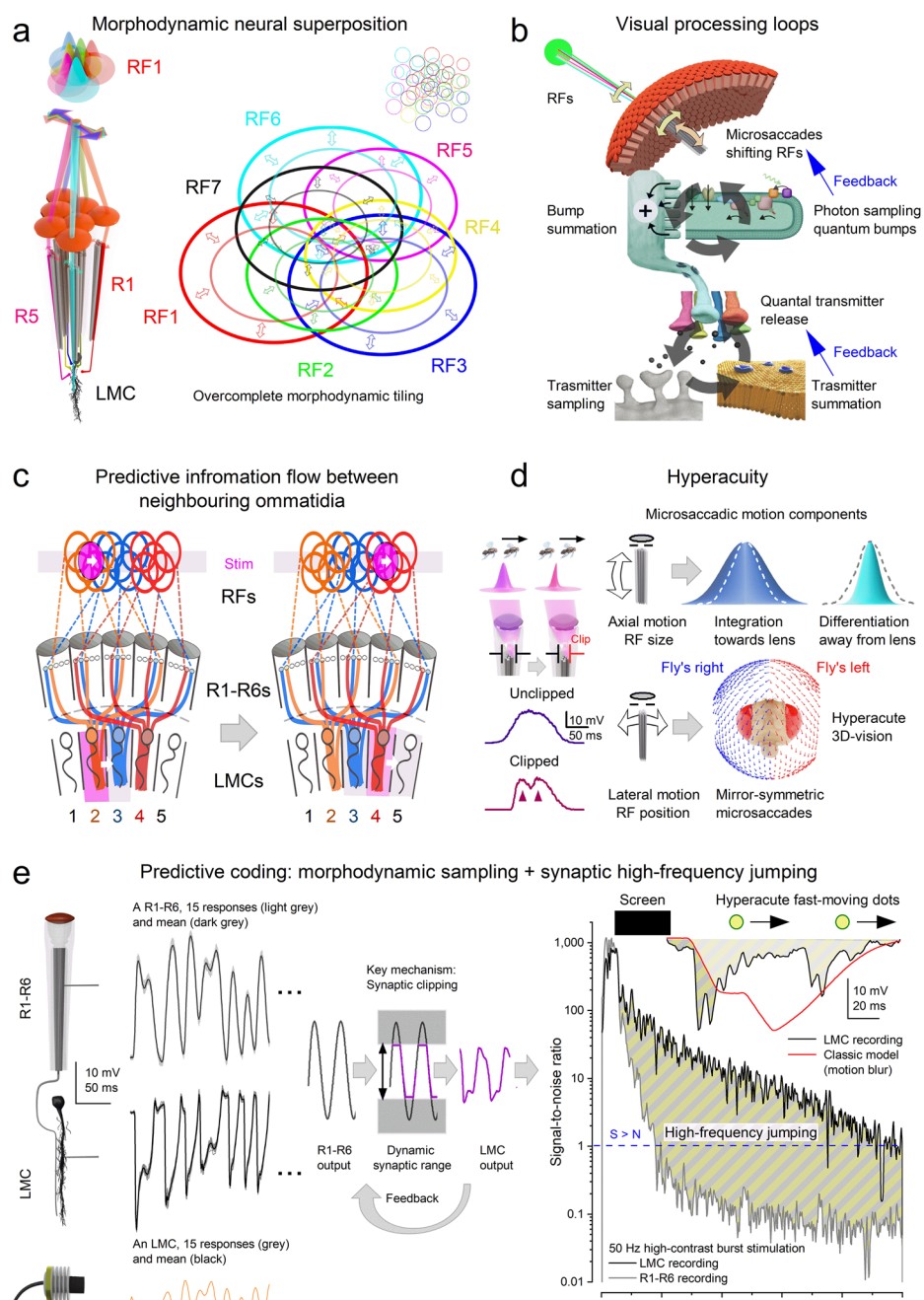

**Fig. 9 | A biophysically realistic morphodynamic neural superposition model of a fly's peripheral vision explains hyperacute predictive coding. a** Overcomplete morphodynamic wiring. R1-R7/8 photoreceptors from neighbouring ommatidia sample light through a "flower-like" arrangement of partially overlapping receptive fields (RFs), collectively feeding a single LMC[11,34]. The LMC receptive field represents the combined morphodynamic sampling matrix, centred on the R7 receptive field (RF7), and dynamically shifts and narrows in response to spatiotemporal light patterns detected by photomechanical R1–R7/8 inputs[1,9–11] (see microsaccade components in d). **b** Photomechanical and synaptic feedback loops. Photomechanical feedback (top): light-driven microsaccades shift RFs in response to local contrast, driving stochastic quantum bump sequences and adapting RF position and size. Synaptic feedback (bottom): voltage differences between R1–R6 photoreceptors modulate quantal histamine release[54–59] onto LMC chloride channels, generating hyperpolarising responses that, in turn, drive excitatory feedback to photoreceptors, balancing synaptic loads and enabling fast, phasic transmission. **c** Predictive lateral signal spread. Because of overcomplete LMC receptive fields (**a**), activation by a moving object (purple disk) simultaneously stimulates neighbouring LMC inputs, producing a laterally propagating, stimulus-locked wavefront. In

addition, mechanical coupling within an ommatidium allows a microsaccade in one photoreceptor to shift a neighbour into the light path[1,9–11], activating adjacent LMCs that would otherwise miss the signal. Together, these mechanisms generate predictive lateral spread. **d** Morphodynamics underlying hyperacuity. Lateral microsaccades can cause aperture clipping (formed by cone and pigment cells[1,9,11]), narrowing receptive fields and enhancing resolution. Similar enhancement occurs under object occlusion (Fig. 6). Additional gains arise when rhabdomeres move inward, collecting light over a narrower angle[10,11]. Mirror-symmetric microsaccades in the two eyes further improve resolvability, enabling hyperacute stereovision[10,11]. **e** Synaptic high-frequency jumping. During high-contrast or saccadic stimulation, photoreceptor responses can exceed the adaptive synaptic range, causing partial clipping. Synaptic high-frequency jumping redistributes power to higher frequencies, extending bandwidth toward ~1000 Hz and producing phasic LMC responses with negligible delay that can peak before photoreceptor signals. Inset: morphodynamic refractory quantal sampling and frequency jumping support predictive coding and motion-blur reduction in natural 3D scenes. Classic static filter models produce blurred responses and fail to reproduce this performance.

*How does behaviour shape neural synchrony to enable ultrafast, predictive sensorimotor processing?* During wakefulness, and especially during active behaviour, flies exhibit heightened visual responsiveness[66,69,73]. Photoreceptor-LMC synapses operate tonically, maintaining continuous interactions between bottom-up sensory input and top-down modulation to support attentional readiness[16–18,53,74]. This dynamic state allows the generation of widespread time-locked neural responses in reaction to behaviourally relevant environmental stimuli.

To understand how high-frequency jumping supports predictive sensorimotor processing, we examined the timing of photoreceptor and LMC responses to rapid stimuli. For example, *Musca* LMCs generate maximum responses within 6.5–9 ms of light onset (mean: $7.6 \pm 0.8$ ms, $n = 10$) - well before the associated photoreceptor voltage reaches its peak 5–9.5 ms later ($p = 0.012$), at 9–16 ms (mean: $11.6 \pm 1.9$ ms, $n = 20$, Supplementary Fig. 1a). Similarly, the fastest reaction times during voluntary vision-driven behaviours fall within 13–20 ms (Fig. 8; Supplementary Fig. 1c), far shorter than expected under conventional unidirectional transmission models. These findings suggest that high-frequency jumping supports both local synaptic efficiency and global network synchronisation, enabling ultrafast, predictive sensorimotor responses.

To assess whether these principles generalise beyond the photoreceptor-LMC synapse, we reexamined response timings deeper in the fly brain. Supporting high-frequency jumping and morphodynamic synchronisation as general neural strategies, minimal-delay responses were also observed downstream in the *Drosophila* visual system[1,66]. In tethered flying flies, electrical activity recorded from the lobula and lobula plate (Supplementary Fig. 1d) - at least three synapses downstream of photoreceptors - appeared within ~15–20 ms of stimulus onset[66], closely time-locked to *Drosophila* LMC transients (Fig. 7c; Supplementary Fig. 1b). Likewise, an $18 \pm 1.5$ ms (mean $\pm$ SD) delay was recorded in the firing of *Drosophila* giant fibres - large command interneurons, four synapses downstream of photoreceptors and involved in collision-avoidance reactions - in response to light-off stimuli[75,76]. This ultrafast signal propagation indicates that latency does not accumulate strictly in proportion to synapse number. Instead, transient light changes evoke near-synchronous activity across successive stages of the optic lobe. These observations challenge classical models that assume slow, strictly sequential processing with substantial cumulative phototransduction and synaptic delays.

Thus, information processing in vivo appears more synchronised and integrated, with signals coordinated across multiple brain regions. This is reflected in the fly brain's broadly distributed and dynamic energy use during activity[77]. Rather than conveying information sequentially like falling dominoes, neurons are coupled through morphodynamic and bidirectional synaptic interactions - with high-frequency jumping providing interlinked "strings" that cause the dominoes to fall together. Such synchronised, minimal-delay processing - from sensing to decision-making - is likely essential for supporting complex behaviours in real time.

*How do neural systems encode space through time to support accurate perception during rapid self-motion?* Analogous to recent concepts of human eye movements[78–80], synaptic high-frequency jumping dynamically shifts neural processing of saccadic inputs into higher-frequency domains, enhancing predictive power and visual acuity. Our findings support this broader framework of encoding space through time[9,11,45,78–83]. Specifically, intracellular recordings and biophysically realistic modelling demonstrate how neural circuits actively transform transient visual signals - such as those elicited by saccades - to synchronise perception precisely with high-speed behaviours. This highlights a conserved principle of dynamic, spatiotemporal encoding across diverse visual systems.

Recent studies suggest that synaptic transmission involves ultrastructural mechanical movements[1,5–8,14,15]. Building on this, we propose

that high-frequency jumping may be sensitised by stochastic ultra-structural oscillations - morphodynamic jitter - driven by tonic transmitter release. This process may help maintain neural processing and perception in an attentive, ready state at synapses transmitting both bottom-up and top-down signals[1]. Morphodynamic jitter, a form of mechanical stochastic resonance[84,85], could enable interconnected circuits to respond in phase (i.e., synchronise) to dynamic or behaviourally relevant inputs, selectively amplifying salient signals while suppressing irrelevant ones. Additionally, jitter could temporally align bottom-up sensory signals with top-down motor predictions[86–88], facilitating faster error correction and behavioural adaptation.

This study underscores the value of an integrative, multi-scale approach to understanding neural systems. By linking molecular, cellular, and systems-level dynamics with high-speed saccadic behaviour, we demonstrate how form, function, and behaviour co-adapt to support robust, adaptive vision in rapidly changing environments - enabling advanced computations already at the level of information sampling. This framework reveals emergent properties - such as high-frequency jumping, efficient coding, hyperacute vision, fast adaptive gain control, and predictive time-locking - that remain obscured when neural components are studied in isolation. For instance, morphodynamic sampling (encompassing photomechanical, stochastic, refractory, and quantal processes) and high-frequency jumping cannot be reproduced by conventional high-level reductionist models that treat photoreceptors and LMCs as static, unidirectional filters.

Beyond insect vision, our findings point to fundamental principles of neural computation. They offer new insights into enduring challenges, such as the neural binding problem, by showing how distributed, time-sensitive signals can be synchronised to generate unified percepts and high-speed, purposeful behaviour. More broadly, these principles could inform the design of next-generation artificial systems that, like biological vision, must operate efficiently under real-time constraints in noisy and dynamic environments.

Looking ahead, uncovering how morphodynamic high-frequency jumping generalises across sensory modalities and species may reveal fundamental laws of biological intelligence - laws that could drive the next revolution in adaptive, real-time artificial systems, from autonomous robots to predictive neuromorphic architectures.

To summarise, our results demonstrate that correctly explaining key performance limits of neural computation - including low effective synaptic delays, wide dynamic range[40,41,45,48], and hyperacute spatiotemporal resolution[1,9,11] - requires models that incorporate biological details usually abstracted away: quantal stochastic sampling, refractoriness and latency variability, ultrastructural motion, and naturalistic input statistics.

Using fly vision as an explicit case, we integrated experimental observations with biophysically realistic modelling of phototransduction and the photoreceptor-LMC synapse. In microvillar photoreceptors, vision is built from tens of thousands of discrete sampling units, where each absorbed photon triggers a stochastic quantum bump and transiently reduces local availability through refractoriness. Early cascade events also produce ultrafast photoreceptor microsaccades[2,9–11,35], coupling ultrastructural motion directly to the timing and statistics of quantal responses. When driven by the bursty contrast dynamics characteristic of naturalistic behaviour, this quantal-stochastic-mechanical substrate naturally produces robust normalisation and gain control (supporting large dynamic range[1,9,45,48]) while sharpening timing relationships that enable hyperacute resolution. Extending the same mechanistic framework to synaptic transmission resolves a long-standing speed puzzle: under naturalistic contrast bursts, the photoreceptor-LMC synapse exhibits *high-frequency jumping*, redistributing transmission toward higher frequencies such that postsynaptic signals become effectively near delay-free, with bandwidths reaching ~1000 Hz and information transfer rates at thousands of bits·s$^{-1}$.

Phenomenological filter-plus-noise or fixed-bandwidth descriptions can often be tuned to match selected response statistics, but they typically do so by imposing auxiliary assumptions - such as externally specified noise models, bandwidth caps, or efficiency constraints - rather than deriving speed, dynamic range, and hyperacuity from identified biophysical mechanisms. Here, these properties instead emerge as coupled consequences of quantal-stochastic-refractory sampling, ultrastructural morphodynamics and synaptic high-frequency jumping under naturalistic inputs, providing a direct mechanistic route from cellular microstructure to behaviourally relevant performance (Fig. 9).

## Methods

We provide here a brief overview of the main methods; full details of the multiscale experimental and theoretical approaches are in the Supplementary Information (Supplementary Notes I–VI). Supplementary Note I covers analyses of intracellular voltage responses from R1-R6 photoreceptors and LMCs under extended experimental paradigms (Supplementary Figs. 1–18 and Supplementary Tables 1–9). Supplementary Note II describes high-resolution X-ray and electron microscopy (EM) analyses of *Musca* compound-eye optics and the photomechanical microsaccadic sampling underlying hyperacuity (Supplementary Figs. 19–25 and Supplementary Tables 10–12). Supplementary Note III explains in vivo high-speed optical imaging of photoreceptor microsaccades (Supplementary Fig. 26). Supplementary Note IV details the mathematical modelling of the morphodynamic neural superposition system, incorporating adaptive optics (Supplementary Figs. 27–38 and Supplementary Tables 13–16). Supplementary Note V presents the behavioural experiments (Supplementary Figs. 39–41). Supplementary Note VI describes the functional connectomics of Musca (Supplementary Figs. 42–47 and Supplementary Tables 17–20).

### Fly stocks

Adult wild-type houseflies (*Musca domestica*) were used in the experiments. Housefly larvae and pupae were sourced from a commercial supplier (Blades Biological Ltd, Cowden, Kent, UK). They were cultured in a standard laboratory incubator (60% humidity) at the School of Biosciences and fed liver and sugar water. Flies were maintained at -22 °C under a 12:12 h light:dark cycle. In some experiments, adult wild-type *Drosophila* (Canton-S) reared separately at 25 °C served as controls[9].

### In vivo intracellular recordings

Fly preparation and intracellular recordings were performed as described previously[9,89]. Briefly, houseflies were anaesthetised on ice. Once immobilised, their wings and legs were removed, and they were fixed to a conical holder (brass/plastic) using beeswax, securing the thorax, proboscis, and right eye to minimise movement artefacts. A small hole (covering 6–10 ommatidia) was cut in the dorsal cornea of the left eye to allow electrode access, and sealed with Vaseline to prevent drying[89].

Voltage responses of R1-R6 photoreceptors and L1-L3 lamina monopolar cells (LMCs) were recorded using sharp, filamented borosilicate microelectrodes (Sutter Instruments; 1.0 mm outer diameter, 0.5 mm inner diameter), with resistances of 100–250 MΩ, pulled with a P-2000 horizontal laser micropipette puller. The tip of the reference electrode was cracked to reduce its resistance.

Photoreceptors and LMCs were recorded in separate sessions. Electrodes were back-filled with 3 M KCl for photoreceptors, and 3 M potassium acetate with 0.5 mM KCl for LMCs to maintain the chloride battery. The reference electrodes (blunt-tipped) were filled with fly Ringer solution (120 mM NaCl, 5 mM KCl, 5 mM TES, 1.5 mM CaCl$_2$, 4 mM MgCl$_2$, and 30 mM sucrose)[40]. Under a Nikon SMZ645 stereomicroscope, a remote-controlled micromanipulator

(PM10, Mertzhauser) was used to position the electrodes. Thanks to the system's stability, a single recording electrode often penetrated multiple photoreceptors - or occasionally LMCs - in sequence, yielding high-quality recordings from many cells within the same eye.

The fly's temperature was maintained at $25 \pm 1$ °C using a feedback-controlled Peltier device[40,89]. Only stable, high-quality recordings were analysed. In darkness, R1-R6 resting potentials were <−60 mV, with responses ≥45 mV to 100 ms saturating light pulses. L1-L3 cells showed dark resting potentials <−30 mV and maximum responses ≥20 mV. As LMCs were blindly penetrated and not stained, individual cell identities could not be confirmed; however, most were likely L1 or L2 due to their larger size[16–18]. Data from all recorded LMCs were pooled, given their similar response properties, including dark resting potential, hyperpolarisation to light increments, and response amplitudes.

### Ruling out efference copy interference in intracellular recordings

Stable intracellular recordings require firmly head-fixed flies to eliminate movement artefacts caused by bodily functions and to minimise recording noise. As a result, the fly cannot perform overt visual behaviours, such as head or body saccades, during stimulation. The saccadic bursty stimuli used in this study, therefore, only mimicked the light intensity time-series patterns (see *Visual stimuli* below) that such behaviours would normally generate. However, the fly could still produce internal motor-related signals - *efference copies* - typically associated with these behaviours[70,86–88].

In our recording paradigm, the fly has no control over the stimulus. Thus, any efference copies generated by its motor circuits and transmitted downstream to photoreceptors or LMCs would not be synchronised with, or predictive of, the stimulus-driven responses. Instead, any such top-down signals would appear as sporadic, uncorrelated noise in the recordings.

We have refined our bespoke intracellular recording systems to achieve exceptionally low noise levels[9,16,40,41,45,65,89]. This enables us to accurately quantify neural responses and distinguish signal from noise under various visual conditions. Our signal-to-noise analyses of photoreceptor and LMC responses (see *Data analysis* below) have never revealed extrinsic noise patterns consistent with efference copies.

Indeed, all observed photoreceptor noise (variability in processing) can be fully explained and reconstructed from four well-characterised sources (see Supplementary Note IV):

- *Quantum bump noise*[40,41], modelled as a Lorentzian derived from the Gamma-shaped bump profile
- *Microsaccade noise*[9], introducing a low-frequency hump
- *Synaptic feedback noise*[18] from LMCs and amacrine cells, adding high-frequency Poisson-like variability shaped by the LMC waveform
- *1/f instrumental recording noise*[40,41]

Given this comprehensive account of known noise sources, the likelihood that efference copies influenced our recorded data under this specific experimental paradigm is exceptionally small. Moreover, our morphodynamic neural superposition model - lacking any built-in top-down circuitry - replicates both photoreceptor and LMC response dynamics (e.g. Fig. 5). Therefore, the present study does not provide evidence for efference copy influence on neural signalling at this early sensory sampling stage.

### Visual point-source stimuli

A high-intensity "white" LED (Seoul Z-Power P4 Star, 100 lumens) was used to deliver light stimuli centred on the receptive field via a randomised quartz fibre-optic bundle (180–1200 nm transmission range) mounted on a rotatable Cardan-arm, subtending a 3° homogeneous

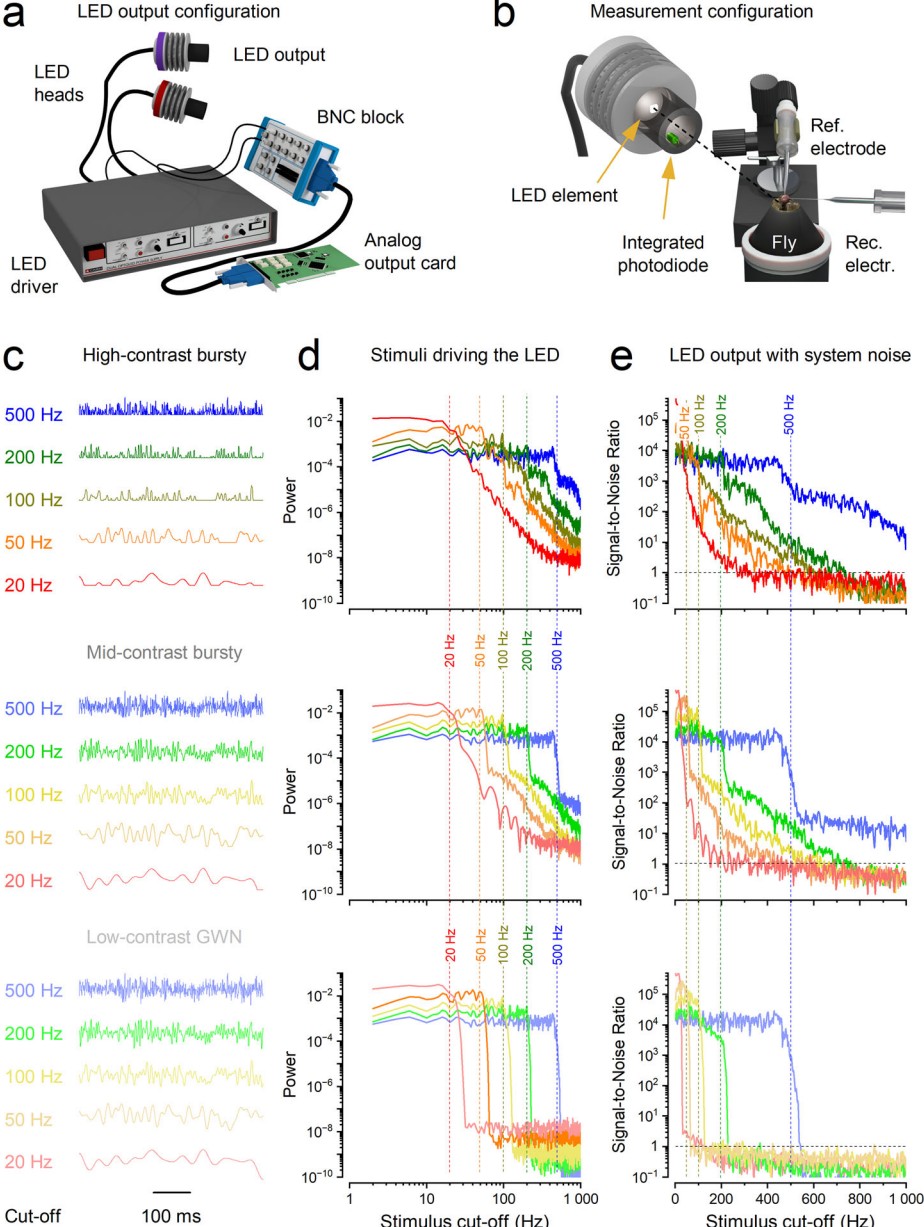

**Fig. 10 | Generation, measurement, and signal-to-noise characterisation of Gaussian white-noise and bursty visual stimuli. a** LED output configuration used for visual stimulation. A high-intensity white LED was driven by an OptoLED controller via an analog output card and BNC block, delivering temporally modulated light through fibre-optic LED heads. **b** Measurement configuration. LED output was monitored directly using an integrated photodiode positioned at the fibre output, while intracellular recordings were made simultaneously from the fly photoreceptor or LMC using sharp microelectrodes and a reference electrode. This configuration allowed direct comparison between commanded stimulus waveforms and measured light output. **c** Example time-domain waveforms of the three stimulus classes used: high-contrast bursty ("saccadic") stimuli (top), mid-contrast bursty stimuli (middle), and low-contrast Gaussian white noise (GWN; bottom). For each class, stimuli were generated with 3 dB cut-off frequencies of 20, 50, 100, 200, and 500 Hz. All stimuli were scaled to equal peak-to-peak modulation and presented for 2 s. **d** Power spectra of the stimuli driving the LED for each contrast class and cut-off frequency, showing flat spectra up to the specified 3 dB cut-off (vertical dashed lines) and rapid attenuation at higher frequencies. Colours correspond to stimulus cut-off frequencies as indicated. **e** Signal-to-noise ratios (SNRs) of the measured LED output for each stimulus condition. Dashed vertical lines indicate nominal stimulus cut-off frequencies. Across all conditions, SNR remained high well beyond the 3 dB cut-off for bursty stimuli, confirming accurate temporal delivery of high-frequency components and defining the usable stimulus bandwidth for subsequent neural analyses.

light field[89]. Output was controlled by an OptoLED driver (Cairn Research Ltd, UK).

To characterise temporal encoding, five Gaussian white noise (GWN) stimuli were presented with varying 3 dB cut-off frequencies (20, 50, 100, 200, and 500 Hz; Fig. 10). Stimuli were 2 s long and sampled at 1000 Hz, or 2000 Hz where noted. Stimuli were generated using MATLAB's *randn* function, low-pass filtered (MATLAB Filter Toolbox), and scaled to have flat power spectra and equal peak-to-peak modulation (two units). Each bandwidth was tested under three contrast conditions on a linear intensity scale: high (BG0, 0 background light units), mid (BG0.5, 0.5 units), and low (BG1, 1 unit). To capture the widest range of stimulus dynamics that *Musca* photoreceptors and LMCs may encounter during fast acrobatic flight, a subset of cells was additionally tested with higher-

bandwidth stimuli with 3 dB cut-offs at 300, 600, and 750 Hz (Supplementary Fig. 11).

Contrast was defined using Weber's law:

$$c = \frac{\Delta I}{I} \tag{1}$$

where $\Delta I$ is the intensity change (standard deviation of the stimulus), and $I$ is the mean background intensity. Measured contrasts were:

- High-contrast "saccadic" bursts: $c(BG0) = 1.29 \pm 0.13$
- Mid-contrast bursts: $c(BG0.5) = 0.61 \pm 0.10$
- Low-contrast GWN: $c(BG1) = 0.33 \pm 0.05$

Stimuli were presented from the lowest to the highest adapting background. Prior to each stimulus, cells were dark-adapted for 20–30 s. Only cells with stable recordings across all 15 stimulus patterns were analysed. However, because LMC recordings are more difficult to maintain than photoreceptor recordings, in some cases only the five bursty stimuli were used for LMCs. Each stimulus was repeated at least 30 times per cell.

Stimuli and responses were low-pass filtered at 500 Hz or 1000 Hz (KEMO VBF/23 elliptic filter, UK) and digitised at 1000 or 2000 Hz using a 12-bit A/D converter (National Instruments, USA). Data acquisition and stimulus control were handled via custom-written software (Biosyst, M. Juusola, 1997–2020) in MATLAB (MathWorks, USA)[40,46], interfaced via the MATDAQ package (H.P.C. Robinson, 1997-2005) for National Instruments boards.

In a series of previous studies[9,44–46,48,49,90,91], we systematically examined how fly photoreceptors encode naturalistic visual signals by directly comparing intracellular voltage responses and information transfer rates across multiple stimulus classes. These included: (i) linearly scanned light-intensity sequences extracted from natural images; (ii) saccadically scanned intensity sequences derived from natural images using measured body movements of freely walking flies[9,23]; (iii) randomly scanned intensity sequences from the same images; (iv) bandwidth-limited Gaussian white noise (GWN) stimuli with 3 dB cut-off frequencies spanning 20–500 Hz; (v) mid- and high-contrast bursty stimuli; and (vi) naturalistic light-intensity time series recorded directly from outdoor environments using van Hateren's calibrated photometric dataset[71].

Across these comparisons, photoreceptors consistently exhibited the highest encoding efficiency, signal-to-noise ratios, and information transfer rates when driven by high-contrast, bursty stimuli that reproduce the rapid, intermittent intensity transients generated during saccadic movements. In contrast, linearly scanned natural images, randomised scans, and GWN - despite having matched or higher mean intensity and comparable nominal bandwidth - elicited weaker responses and lower information throughput.

These experimental findings were further validated using a biophysically realistic stochastic, refractory, quantal photoreceptor model, which reproduced the same performance advantages under bursty stimulation. In the present study, we extend this experimentally validated modelling framework from *Drosophila* photoreceptors (which contain ~30,000 microvilli) to *Musca* photoreceptors (~54,000 microvilli), while preserving the same photomechanical phototransduction principles and stimulus-response relationships.

Thus, the saccadic light stimuli (mid- and high-contrast bursts) used here are not ad hoc abstractions but are grounded in direct comparisons with natural image statistics, measured fly behaviour, and independently validated biophysical models, ensuring that they accurately capture and adequately span the range of rapid intensity dynamics experienced during natural saccadic vision.

## Visual moving grating and dot stimulation

The moving visual-field and moving object stimulus methods and protocols, including the use of occlusions to clip the R1-R6s' receptive fields to test photoreceptors' and LMCs' visual acuity, are explained in Supplementary Notes I.7-8 (Supplementary Figs. 14–18 and 24, 25).

## Data analysis

To ensure all the studied cells had reached a similar adaptation state, the first five responses (10 s of data) to the repeated stimulus were excluded from both signal and noise analyses. This left at least 25 responses to the same repeated stimulus pattern. The signal was defined as the mean response, and the noise as the deviation of individual traces from this mean[40,46]. Thus, $n$ repetitions ($n = 30$) yielded one signal and 25 noise traces.

The signal s(t) and noise n(t) traces were segmented into 50%-overlapping stretches and windowed with a Blackman-Harris 4-term window. Each window produced six 500-point- or 1000-point-long samples, corresponding to 1 or 2 kHz sampling, respectively. Fast Fourier transforms (FFTs) were applied to compute the frequency-domain signal and noise spectra, S(f) and N(f), respectively. The signal-to-noise ratio in the domain SNR(f) was calculated as:

$$SNR(f) = \frac{|\langle S(f)\rangle|^2}{|\langle N(f)\rangle|^2} \tag{2}$$

where $|\langle S(f)\rangle|^2$ and $|\langle N(f)\rangle|^2$ are the power spectra of signal and noise, respectively. Here $v$ denotes voltage, $||$ the absolute value, and $\langle\rangle$ the average over all signal and noise windows[40,46].

Information transfer rate (R) was calculated from the SNR(f) using Shannon's information formula[64], which is widely applied in this context[9,44,46]:

$$R = \int_{f_{low}}^{f_{high}} \log_2(SNR(f)+1)df \tag{3}$$

Signals were sampled at either 1 or 2 kHz and windowed accordingly (1000- or 2000-point Blackman-Harris window). Therefore, the integration bounds were 2–500 Hz (for 1 kHz sampling) or 1–1000 Hz (for 2 kHz sampling), not 0 to ∞.

However, for LMC recordings sampled at the lower rate (1 kHz), Eq. 3 underestimates the true information transfer rates—particularly for mid- and high-contrast "saccadic" burst responses—because these evoked high-frequency jumping, with SNR(f)≫1 at 500 Hz (*cf*. Fig. 3e), indicating that frequencies above 500 Hz contributed non-negligible information. In contrast, this underestimation was not observed for responses to low-contrast Gaussian white noise (GWN) stimuli, which did not evoke high-frequency jumping and exhibited substantially lower response bandwidths (Supplementary Fig. 7e).

To correct for the high-frequency jumping effect, information losses in the 1 kHz recordings were estimated by comparison with matched 2 kHz recordings using the same stimuli. For photoreceptors ($n = 2$), the mean information loss was approximately 5% and consistent across cells. For LMCs ($n = 2$), the loss ranged from 5–23%, with the largest deficits observed for stimuli peaking near 200 Hz. The information transfer rate estimates for both photoreceptors and LMCs were corrected using stimulus-specific factors, calculated as the percentage difference between $R_{2-500Hz}$ and $R_{2-1,000Hz}$ as defined by Eq. 3.

Moreover, information transfer rate estimates for LMCs during high-contrast bursts were less reliable than those for photoreceptors because the recorded voltage signals typically deviated from a Gaussian distribution—except when tested with a 500 Hz Gaussian White Noise stimulus (Supplementary Note I, Supplementary Fig. 4b). Consequently, Shannon-based information estimates are most accurate under mid- and low-contrast conditions, where voltage responses more closely approximate Gaussian distributions (Supplementary

Fig. 7b). Applying Shannon's method to non-Gaussian responses, which violates its assumptions, may inflate estimates by ~12%, as verified against the assumption-free triple-extrapolation method[46] (Supplementary Fig. 8).

However, several factors contribute to underestimating the true capacity of the system (Supplementary Fig. 9). Microelectrode penetrations inherently damage the recorded cells, reducing signal fidelity. Additionally, responses cannot be measured in a true steady state, as they continually reflect ongoing adaptation, network dynamics[17,18,63] and top-down eye-muscle activity[92]. Because recordings are never fully ergodic - trial-to-trial variability arises from adaptive network processes rather than pure noise - standard stationary information analyses mistakenly classify intrinsic or network-driven adaptations as additive noise. Thus, the actual information transfer rates and visual performance of housefly vision likely surpass our conservative estimates.

These analyses demonstrate that estimates derived from 25 repetitions (excluding the first five responses) are conservative rather than inflated; adaptive trends cause underestimation, not overestimation, of synaptic information transfer capacity. Therefore, our reported values represent a robust lower bound on the actual encoding precision of LMCs.

Flies also counteract motion blur during saccadic behaviours via multiple mechanisms: predictive stabilisation of head and body movements[25,26], enhanced processing in acute zones[93], and photomechanical, refractory light information sampling[9,11]. However, in our experiments, *Musca* were fixed in conical holders, eliminating head and body movement. We also used female flies and did not intentionally target the male acute zone, ensuring that neither electrode placement nor sex biased the results. Thus, the enhanced responsiveness of both photoreceptors and LMCs to fast, saccade-like stimuli reflects high-performance sampling and transmission dynamics[9–11], which have evolved to support the fly's high-speed visual behaviours and lifestyle.

### Morphodynamic neural superposition system

In the fly's neural superposition eye (as viewed statically), six R1-R6 photoreceptors from different ommatidia converge onto shared downstream neurons (LMCs and an amacrine cell)[94]. Each photoreceptor is optically aligned to sample light from approximately the same region of space - but due to slight angular offsets and biological variability, their receptive fields do not perfectly overlap[11,34]. Instead, they sample from a small, fuzzy area, creating an over-complete, spatially jittered representation of the visual scene (Fig. 2b).

In a living fly, however, this system operates as a spatiotemporally dynamic morphodynamic network[11]. The receptive fields are not fixed; they shift in space and time due to photomechanical microsaccades - tiny, light-driven rhabdomere movements (Fig. 2b, c). This adds a temporal dimension to the over-complete spatial sampling, allowing receptive fields to sweep across fine spatial details and generate richer, decorrelated input patterns.

The result is a morphodynamic neural superposition system that enhances information encoding by:

- Dynamically refining receptive field alignment,
- Exploiting redundancy for noise suppression and error correction,
- Supporting high-frequency jumping and predictive coding aligned with behaviour.

This system transforms passive optical overlap into an active, synchronised sampling strategy, optimised for high-speed saccadic vision. Supplementary Note IV details how we modelled the photoreceptor and LMC responses of this sophisticated morphodynamic system.

### Extrapolating *Musca* reaction times

To estimate the minimal reaction time of *Musca* antennal and leg responses to visual stimuli, we focused on two fast sensorimotor

pathways: the light-induced antennal movement and the looming-induced leg escape response (see Supplementary Note VI, Supplementary Figs. 42–47). Although no complete connectomic or genetic dataset exists for *Musca*, the neural architecture underlying the visual system is highly conserved between *Drosophila* and *Musca*[95]. We therefore used *Drosophila melanogaster*, for which complete connectomic data is available. Using the FlyBrainLab platform (see Supplementary Note VI for details), we identified the shortest reflex-like pathways linking the retina to motoneurons controlling antennal and leg responses (Fig. 7d, h), with minimal synaptic gaps and conduction distances. Examining these pathways in Drosophila allows us to extrapolate *Musca*'s visual-motor response time.

We accumulate processing delays along the two *Drosophila* pathways, attributed to phototransduction and voltage integration, synaptic transmission delay, and conduction speed (see Supplementary Note VI). While *Musca* has a larger brain, the main contributors to processing delay, phototransduction and synaptic transmission, are likely conserved, while a small difference in conduction delays may exist and we consider that the *Musca* has 3 times the body size as that of the fruit fly.

Together, these stages yield an estimated reaction time of 22.5–29.5 ms for the light-induced antennal movement and 18.6–22 ms for the looming induced leg escape response. We propose this as a reasonable lower-bound estimate for these visually induced motor responses in *Musca*, based on highly compact reflex-like pathways and conventional fast signalling assumptions. These estimates are at least 5–9 ms slower than our experimental observations of voluntary visual responses in flies (13 ms), suggesting the involvement of in vivo acceleration mechanisms (e.g., synaptic high-frequency jumping).

### Statistics

Statistical analyses were carried out in Python, Origin and MATLAB. Maximum information rates and visual acuity between male and female photoreceptors, and the whole population of photoreceptors and LMCs, were compared. The behavioural data were tested against statistical models. The statistical methods are explained in the Supplementary Notes I–VI.

### Illustrations and videos

Apart from a small number of hand-drawn illustrations, most three-dimensional visualisations, Supplement Movies, and illustrative artwork were created using the open-source software Blender (Blender Foundation; https://www.blender.org), across multiple software versions.

### Reporting summary

Further information on research design is available in the Nature Portfolio Reporting Summary linked to this article.

## Data availability

The data supporting the findings of this study are available from the corresponding authors upon request.

## Code availability

All the software code, including the complete *Musca* morphodynamic neural superposition model, can be downloaded from: https://github.com/JuusolaLab/High-frequency-jumping and https://github.com/JuusolaLab.

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

## Acknowledgements

We thank B. Webb, R.C. Hardie, A. Nikolaev, A. Lin and J. Stone for discussions and comments, members of the Juusola laboratory for discussions, and the Duke X-ray imaging team and staff at EMBL Hamburg for their assistance. SBFSEM data of *Musca* ommatidia was generated in the Oxford Brookes University Centre for Bioimaging. This work was supported by Jane and Aatos Erkko Foundation Fellowships (MJ and JT), The Leverhulme Trust (RPG-2024-016: MJ and PV), the Biotechnology and Biological Sciences Research Council (BB/F012071/1, BB/D001900/1, and BB/H013849/1: MJ), the Engineering and Physical Sciences Research Council (EP/P006094/1: MJ) and Horizon Europe Framework Programme grant NimbleAI - Ultra energy-efficient and secure neuro-morphic sensing and processing at the endpoint (LC).

## Author contributions

Conceptualisation (M.J., J.T., and N.M.), Investigation (N.M., J.T., M.J., J.K., A.D.B., H.M., A.A.B., K.A., T.R., B.Y.B., S.S., Y.Z., M.K., J.O., J.M., M.L. and P.V.), Methodology (M.J., J.T., J.K., M.K., A.A.L., Y.Z., E.D., P.V.), Project administration (M.J., E.D., A.A.L., P.V., L.C.), Resources (M.J., P.V., A.A.L., L.C.), Software (J.T., J.K., H.M., B.B., S.S., Y.Z., A.A.L., and M.J.), Supervision (M.J.), Writing - main paper original draft (M.J.), Writing - review & editing (M.J., J.T., A.D.B., J.K., H.M., V.V., M.K., A.A.L., Y.Z., P.V., M.L., M.K., G.d.P., E.D., L.C. and N.M.).

## Competing interests

The authors declare no competing interests.
