## [Transparent Peer Review file · Nature Communications]

Synaptic high-frequency jumping synchronises vision to high-speed behaviour

Corresponding Author: Professor Mikko Juusola

Version 0:

Reviewer comments:

Reviewer #1

(Remarks to the Author)

This manuscript uses the housefly visual systems and state-of-the-art technologies to investigate the mechanisms that allow neural systems to achieve accuracy during ultrafast motion. It reveals a previously unknown mechanism, synaptic high-frequency jumping, in which photoreceptor-LMC synapses dynamically shift transmission towards higher frequencies during saccadic input. This manuscript also explains how this mechanism emerges and describes visually triggered behaviors with extraordinarily brief delays.

The following minor suggestions aim to improve the manuscript's clarity and help readers better understand the work.

1. One point that is difficult to understand is that two different parameters share the same unit (Hz): the stimulus temporal frequency and the bandwidth. While the meaning of stimulus temporal frequency is clear, it is less obvious how bandwidth is defined when it is used to describe photoreceptors and LMCs. In addition, what is the relationship between the "information rate (bits/s)" and "bandwidth"?
2. Page 6 (line 5) mentions Fig 2b. It appears that this description corresponds to Fig. 2c.
3. Page 6 (lines 9-16) describes the drawbacks of GWN stimuli. It would be helpful to clarify the differences between bursty stimuli and GWN stimuli, and to explain how bursty stimuli address these limitations. Specifically, in Fig. 2c, do the light stimulus patterns (high-contrast bursty, mid-contrast bursty, and low-contrast GWN) differ only in contrast, or also in their temporal structure and other parameters?
4. Page 9 (line 1) says that "the biphasic nature of LMC responses effectively doubled....." The photoreceptor limit is ~230 Hz, and the LMC limit is ~920 Hz. It seems that the increase is closer to fourfold rather than twofold. Could the authors clarify why this is described as a doubling rather than a quadrupling?
5. Page 9 (lines 14-15) says that authors "used ten saccadic light patterns and five randomised light patterns (Gaussian white noise, GWN) as controls." From Figures 3a, 3b, and 2c, it is difficult to discern or interpret the specific light stimulus patterns presented.
6. Figure 3d, the x-axis label has a typo. The word should be bandwidth, not bandwidth.
7. Figure 3 shows that speeding up bursty stimulus increases the photoreceptor bandwidth. It does not further enhance the LMC bandwidth. Is it expected? Could the authors provide an explanation or comment on this observation?
8. Page 14 (line 37): Figure 4g does not support the claim in the sentence "removing excitatory feedback from lamina interneurons to photoreceptor terminals substantially degrades both photoreceptor and LMC signal fidelity, consistent with experimental findings."
9. In Supplementary Video 1, why are the representative responses shown at 45 s and 75 s from an interneuron rather than from an LMC?
10. Page 15 (line 38) should be Figure 5f, not Figure 5d.
11. Page 16 (line 7) should be Figure 5c, not Figure 5b.
12. Inconsistent synopsis number between Figure 6d and Page 17 (line 16).
13. Page 17 line 23 mentions Figure 6h-i, but there is no Figure 6i in Figure 6.

(Remarks on code availability)

Reviewer #2

(Remarks to the Author)

This study investigates how the *Musca domestica* visual system achieves ultrafast information transfer during rapid, saccadic behaviours. Using a combination of electrophysiology, optical imaging, electron microscopy, and computational modeling, the authors identify a previously undescribed mechanism “synaptic high-frequency jumping” by which photoreceptor–LMC synapses dynamically shift visual information into higher-frequency carrier bands. They propose that this process enables predictive, near-instantaneous visual encoding that aligns with voluntary behavioural responses occurring within 13–20 ms. Although the study is technically ambitious and addresses a topic of considerable interest, the current manuscript exhibits significant conceptual, methodological, and analytical shortcomings. Specifically, the principal assertions lack robust quantitative support, interpretations often extend beyond the available evidence, and several essential analyses and control experiments are absent.

Major comments

1. There is still insufficient evidence to demonstrate that high-frequency jumping in the photoreceptor–LMC circuit effectively reduces motion blur during flight in houseflies.
2. The primary assertion that synaptic high-frequency jumping facilitates predictive coding is not rigorously supported by quantitative analysis. The study does not include direct measurements of prediction error, comparative model analyses, or information-theoretic evaluations. Consequently, the hypothesis that “synaptic high-frequency jumping” underlies predictive coding remains primarily descriptive. The connection to predictive coding appears metaphorical rather than empirically demonstrated, with no quantitative data indicating prediction error minimization or feedback-mediated suppression. To substantiate this claim, model-based comparisons between expected and observed signals should be incorporated to demonstrate that high-frequency jumping effectively reduces temporal prediction error.
3. The authors should clarify how the saccadic light stimuli used in this study accurately mimic the rapid intensity changes occurring in natural environments and describe how the stimulation parameters were determined or validated.
4. The data in Fig. 3e–f show that photoreceptors respond most strongly to 200 Hz light stimulation, whereas LMCs, following high-frequency jumping, respond optimally around 100 Hz. This raises important questions about the functional link between high-frequency jumping and motion-blur reduction. The authors should clarify this relationship and provide additional evidence to support the proposed link between high-frequency jumping and temporal precision in visual processing. Exploring the upper frequency limit at which photoreceptors or LMCs can reliably respond to bursty light stimuli, and assessing how different physiological or environmental conditions (e.g., Fig. 4c–f) affect this responsiveness, would provide deeper insight into the mechanism’s biological relevance during high-speed flight.
5. The signal-to-noise ratio (SNR) analyses employ an arbitrary threshold ($\text{SNR} > 1$) without justification. The rationale for this criterion should be clarified, or alternative thresholds (e.g., 3 dB) should be used to test the robustness and reliability of the results.
6. The manuscript frequently refers to concepts such as “thermodynamic constraints” and “energy-efficient coding,” implying that the proposed mechanism reduces metabolic cost. However, no empirical measurements of metabolic parameters such as ATP consumption, ionic current quantification, or oxygen utilization are provided to substantiate these claims.
7. The comparison between high-contrast saccadic bursts and Gaussian white noise (GWN) is central to the study, yet these stimuli differ in multiple aspects (temporal structure, mean luminance, and contrast adaptation). It therefore remains unclear whether the observed effects arise from “naturalistic structure” or simply from higher contrast levels.
8. The morphodynamic model appears complex and may involve numerous fitted parameters. The authors should clarify which parameters were empirically constrained and provide sensitivity or robustness analyses to validate the model’s reliability.
9. It remains unclear whether the L1/L2 LMC subtypes receive histaminergic input from R1–R6 photoreceptors in houseflies. The authors should investigate whether histamine is released from R1–R6 during high-speed visual input and whether histamine receptors are expressed in L1/L2 LMCs.
10. The observed high-frequency components could potentially arise from synchronized vesicle release or short-term synaptic depression at the histaminergic synapses between R1–R6 and LMCs. The authors should consider and discuss these alternative mechanisms.
11. The authors use *Drosophila* connectome data to infer visual circuit connectivity and transmission speed in *Musca domestica* (Fig. 6). This approach is inadequate given the major differences in brain size and neuron number between the two species. Furthermore, while many supporting studies cited involve *Drosophila*, the present experiments were conducted in *Musca*. The degree to which these two species share comparable neural circuitry and behavioural dynamics remains uncertain and should be explicitly discussed.
12. The antennal and leg reaction times reported for brief UV flashes and high-speed looming stimuli are not compelling. The authors should clarify whether houseflies exhibit other behavioural responses to these stimuli and provide further quantitative data to strengthen their conclusions.

Minor comments

1. Is the definition of high- and mid-contrast bursty stimuli, as well as low-contrast Gaussian white noise in Fig. 2c, based on any previous references or studies?
2. Only one LMC dataset was used for multiscale simulations in Fig. 1c, which appears insufficient.
3. It is recommended to standardize all units throughout the manuscript (e.g., Hz vs. kHz; bits/s vs. bits s⁻¹).
4. Use “photoreceptor–LMC synapse” consistently instead of alternating with “first visual synapse” or “lamina transmission.”
5. Fig. 2–3 require clearer axis labels and inclusion of sample sizes (n). The color coding across different stimuli (20–500 Hz) should be unified across figures and supplementary data.
6. The text states that “the biphasic nature of LMC responses (Fig. 2d) effectively doubled the frequency content of their photoreceptor inputs.” However, if a 230 Hz input is quadrupled to 920 Hz, this appears to be an error.
7. The differences between the light stimulation paradigms used in Fig. 2 and Fig. 3 should be clearly explained, as the text currently does not distinguish them.

Overall Recommendation

The manuscript is excessively long and conceptually diffuse, blending empirical results with theoretical discussion in a way that lacks clear structural progression. In its current form, readers may find it difficult to follow the logical flow from experimental observations to mechanistic interpretation. Furthermore, the term “morphodynamic neural superposition” is repeatedly used without a formal definition. The authors should provide a concise and explicit definition of this concept and explain how it differs from classical neural superposition to clarify how it advances current understanding. The supplementary material is also excessively extensive (over 100 pages) and contains redundant descriptive content that could be condensed or relocated to a dedicated data repository. Streamlining the supplementary section would greatly improve readability and maintain focus. In summary, the manuscript would benefit from substantial revision to enhance clarity, logical coherence, and accessibility for a broad readership.

(Remarks on code availability)

Version 1:

Reviewer comments:

Reviewer #1

(Remarks to the Author)

I am satisfied with the authors' revisions and responses. Only one minor error was noted:
Line 207: Fig. 4a to Fig. 4c.

(Remarks on code availability)

Reviewer #2

(Remarks to the Author)

The authors have addressed most of my previous concerns in this revised manuscript, and the study has been substantially improved. I have only one remaining point for clarification.

In Fig. 6c, the data indicate that houseflies can resolve two points separated by 0.7° at a lower angular velocity (42° s^{-1}), or 1.4° at a higher angular velocity (168° s^{-1}), but fail to resolve 0.7° at the higher angular velocity (168° s^{-1}). The authors propose that high-frequency jumping (HFJ) enhances visual performance in high-speed environments. However, it remains unclear whether the angular velocities used in these experiments correspond to biologically relevant speeds encountered during natural flight. To strengthen the biological relevance of this conclusion, could the authors translate the visual stimulus frequency and corresponding angular velocity into an estimated real-world flight speed of the fly? Such an analysis would allow a more direct comparison with reported average flight speeds of houseflies and help determine whether the experimental conditions indeed represent low- and high-speed visual environments in a biologically relevant context.

(Remarks on code availability)

RESPONSE TO REVIEWER COMMENTS

Dear Editors,

We thank the reviewers for their constructive and insightful feedback, which has substantially strengthened the manuscript. We have carefully considered all suggestions and revised the paper to clarify its findings and improve accessibility for a broad readership.

Below, we provide point-by-point responses to each reviewer comment (numbered in **blue**). In a separate PDF containing the revised main manuscript and Supplementary Information, all corresponding changes are highlighted in **red**.

In response to the reviewers' comments, we have introduced the following major additions and revisions:

- **New introductory Fig. 1**, clarifying and contrasting the morphodynamic neural superposition system with the classical static wiring assumption of the fly compound eye, and illustrating how morphodynamic, refractory, stochastic quantal sampling contributes to predictive coding and motion-blur reduction.
- **New Results Fig. 6 and Supplementary Fig. 18**, comparing intracellular recordings with simulated LMC voltage responses under progressively reduced neural superposition models:
 - Full morphodynamic neural superposition with 42% clipped receptive fields, closely matching recordings.
 - Static neural superposition (no photoreceptor microsaccades) with overcomplete receptive fields, synaptic feedback, and stochastic refractory quantal sampling.
 - Quasi-classical neural superposition (no microsaccades) with perfectly aligned photoreceptor receptive fields of identical size, while retaining stochastic refractory quantal sampling and synaptic feedback.
 - Classic stationary neural superposition filter model following pioneering assumptions, exhibiting the slowest responses and poorest spatiotemporal resolution.
- **New Discussion Fig. 9**, illustrating how the biophysical components of the morphodynamic neural superposition system interact to produce synaptic high-frequency jumping and predictive, hyperacute vision with minimal delay and noise.
- **New Methods Fig. 10**, depicting the time-series point-source stimuli (from Gaussian white noise to high-contrast saccadic bursts) used in this study.
- **New Supplementary Figs. 11–13**, presenting additional experiments using further stimuli, extending to higher 3 dB cut-off frequencies (up to 750 Hz) and spanning a million-fold intensity range from darkness to bright daylight.
- **New Supplementary Figs. 40–41**, presenting statistical analyses of the light-triggered visual behaviours (Antennae moment and startling leg-rise responses).
-

In addition, we have carefully revised the entire manuscript and Supplementary Information, including all figures, to ensure clarity, consistency, and alignment with *Nature Communications* house style.

We hope that, with these revisions, the manuscript is now suitable for publication in *Nature Communications*.

Yours sincerely,

Mikko Juusola

on behalf of all authors

Reviewer #1 (Remarks to the Author):

This manuscript uses the housefly visual systems and state-of-the-art technologies to investigate the mechanisms that allow neural systems to achieve accuracy during ultrafast motion. It reveals a previously unknown mechanism, synaptic high-frequency jumping, in which photoreceptor-LMC synapses dynamically shift transmission towards higher frequencies during saccadic input. This manuscript also explains how this mechanism emerges and describes visually triggered behaviors with extraordinarily brief delays. The following minor suggestions aim to improve the manuscript's clarity and help readers better understand the work.

1. One point that is difficult to understand is that two different parameters share the same unit (Hz): the

stimulus temporal frequency and the bandwidth. While the meaning of stimulus temporal frequency is clear, it is less obvious how bandwidth is defined when it is used to describe photoreceptors and LMCs. In addition, what is the relationship between the “information rate (bits/s)” and “bandwidth”?

Our Reply: We thank the reviewer for highlighting this important point and agree that clearer definitions are required. We have now explicitly distinguished between stimulus temporal frequency, neural encoding bandwidth, and information rate throughout the manuscript.

Specifically, the stimulus temporal frequency is now referred to as the stimulus cut-off frequency, defined by the original 3 dB cut-off used when generating the stimulus waveform. In contrast, the effective photoreceptor and LMC bandwidths refer to an output property of the neural response and are now consistently defined as the highest frequencies at which the response signal-to-noise ratio (SNR) exceeds 1.

The information rate (bits s^{-1}) quantifies the total amount of information transmitted per unit time and depends jointly on both the encoding bandwidth and the frequency-dependent SNR of the response. Thus, while bandwidth (Hz) specifies the range of frequencies over which information can be transmitted, the information rate integrates information across that range and is not directly equivalent to bandwidth.

We have revised the Methods and Results text to reflect these definitions consistently and to clarify the distinction between input stimulus statistics and output neural encoding properties.

2 Page 6 (line 5) mentions **Fig 2b**. It appears that this description corresponds to **Fig. 2c**.

Our Reply: Owing to the addition of a new introductory Fig. 1, the figure referred to in the reviewer’s comment is now Fig. 2. We thank the reviewer for noting this error. The figure reference has now been corrected so that the text correctly refers to Fig. 2c in the revised manuscript.

3 Page 6 (lines 9-16) describes the drawbacks of GWN stimuli. It would be helpful to clarify the differences between bursty stimuli and GWN stimuli, and to explain how bursty stimuli address these limitations. Specifically, in **Fig. 2c**, do the light stimulus patterns (high-contrast bursty, mid-contrast bursty, and low-contrast GWN) differ only in contrast, or also in their temporal structure and other parameters?

Our Reply: We thank the reviewer for this helpful suggestion and agree that a clearer distinction between bursty and GWN stimuli strengthens the manuscript.

We now clarify that the stimuli in (the renumbered) **Fig. 3c** differ not only in contrast, but also fundamentally in their temporal structure and frequency composition. Specifically, bursty stimuli are designed to capture the intermittency, clustering, and high-amplitude transients characteristic of natural visual input during saccadic flight, whereas Gaussian white noise (GWN) is temporally stationary and lacks such structured events.

To make this explicit, we have added a new Methods Figure (**Fig. 10**) that directly compares the frequency content of high-contrast bursty, mid-contrast bursty, and low-contrast GWN stimuli at their respective 3 dB cut-off frequencies. This analysis shows that, even when nominal bandwidth limits are matched, bursty stimuli contain temporally localised events that selectively drive transient high-frequency components, whereas temporally stationary GWN - with its higher mean intensity - engages photoreceptor refractoriness more uniformly and elicits weaker high-frequency responses.

In addition, we have added **Supplementary Figs. 11-13**, together with new **Supplementary Notes**, which demonstrate:

- (1) how synaptic high-frequency jumping reaches the highest effective LMC response bandwidths specifically during 20-300 Hz high-contrast bursts;
- (2) how the magnitude of high-frequency jumping scales with light intensity, revealing a strong interaction between stimulus statistics and photoreceptor operating regime; and
- (3) why low-contrast GWN systematically underestimates achievable synaptic bandwidth and information transfer under naturalistic conditions.

We have also revised the relevant Results text to explicitly distinguish stimulus contrast from stimulus temporal structure. The revised paragraph now reads:

“We performed electrophysiological experiments (**Fig. 3a**) using ‘saccadic’ light stimuli that mimic the rapidly changing, burst-dominated intensity patterns experienced by photoreceptors during natural visual behaviours. These bursty stimuli combined high-contrast transients with temporally clustered fluctuations spanning multiple frequency bands, with 3 dB cut-off frequencies of 20, 50, 100, 200, and 500 Hz (Fig. 3c).

Contrast ranged from moderate ($c \approx 0.6$) to high ($c \approx 1.5$). As a control, and to benchmark our results against classical linear-systems approaches⁴⁷⁻⁵¹, we also applied low-contrast ($c \approx 0.3$), bandwidth-limited Gaussian white noise (GWN) stimuli, which are temporally stationary and lack burst structure (**Fig. 3c**). Our primary focus here is on diurnal conditions, when houseflies are most active. Additional experiments using these and further stimuli - including higher 3 dB cut-off frequencies - tested responses across a six-log-unit range of light intensities, from darkness to bright daylight, and are described in **Supplementary Notes 1**."

Together, these additions clarify that bursty stimuli address the limitations of GWN by engaging nonlinear, intensity-dependent synaptic mechanisms that are underestimated under stationary stimulation, thereby revealing the functional role of synaptic high-frequency jumping during natural high-speed behaviour.

4. Page 9 (line 1) says that "the biphasic nature of LMC responses effectively doubled....." The photoreceptor limit is ~230 Hz, and the LMC limit is ~920 Hz. It seems that the increase is closer to fourfold rather than twofold. Could the authors clarify why this is described as a doubling rather than a quadrupling?
Our Reply: We agree that the original phrasing was imprecise and have corrected it in the revised manuscript. The term "doubled" has now been replaced with language that accurately reflects the measured bandwidth expansion.

Importantly, photoreceptor effective bandwidth is not fixed at the classic flicker-fusion frequency (~230 Hz), but depends on the total number of microvilli (sampling units) within each photoreceptor. In the revised text, we now explicitly state that during fast saccadic stimulation photoreceptor responses can reach effective bandwidths of up to ~440 Hz (signal-to-noise ratio > 1), with variability across recordings reflecting natural differences in microvillar number and rhabdomere geometry (**Figs. 3e** and **4a**; **Supplementary Figs. 20-22**).

LMC responses can extend well beyond this presynaptic range owing to synaptic high-frequency jumping, such that the increase in frequency content exceeds a simple twofold scaling. The revised text and Fig. legends now describe these relationships quantitatively and consistently, resolving the confusion noted by the reviewer. These corrections do not affect the conclusions of the study.

5. Page 9 (lines 14-15) says that authors "used ten saccadic light patterns and five randomised light patterns (Gaussian white noise, GWN) as controls." From **Figs. 3a, 3b**, and **2c**, it is difficult to discern or interpret the specific light stimulus patterns presented.

Our Reply: We agree with the reviewer that, in the original version, it was difficult to discern the specific temporal structure of the different light stimulus patterns from Figs. 2c and 3 alone. To address this, we have added new Figs. that explicitly visualise and quantify the stimulus waveforms and their frequency content.

Specifically, we now include a new Methods Fig. (**Fig. 10**) showing the time-domain waveforms and corresponding frequency spectra for all light stimuli used in the study, including the ten saccadic (bursty) patterns and the five randomised Gaussian white noise (GWN) control patterns. In addition, while **Supplementary Figs. 1-10** already provided representative photoreceptor and LMC response recordings for each stimulus class (low-contrast GWN, mid-contrast bursty, and high-contrast bursty) across the tested stimulus cut-off frequencies (20-500 Hz), we have now added **Supplementary Figs. 11-13**. These new figures present representative photoreceptor and LMC responses to high-contrast bursty stimuli over an extended frequency range (20-750 Hz) and show the dependence of responses to high-contrast bursty, mid-contrast bursty, and GWN stimuli on light intensity across a million-fold range (six log units).

Together, these additions allow readers to directly compare stimulus structure, spectral content, and neural responses across conditions, making the distinctions between saccadic and GWN stimuli explicit and facilitating interpretation of the corresponding results. We now explicitly refer to these figures in the main text at the relevant locations.

6. **Fig. 3d**, the x-axis label has a typo. The word should be bandwidth, not bandwidth.

Our Reply: corrected.

7. **Fig. 3** shows that speeding up bursty stimulus increases the photoreceptor bandwidth. It does not further enhance the LMC bandwidth. Is it expected? Could the authors provide an explanation or comment on this observation?

Our Reply: Yes, this behaviour is expected and is a direct consequence of the mechanism underlying synaptic high-frequency jumping.

High-frequency jumping arises primarily from dynamic thresholding (signal clipping) at the photoreceptor-LMC synapse. When bursty stimuli drive sufficiently large and rapid photoreceptor voltage excursions, the synaptic operating range is exceeded, and the resulting nonlinear clipping redistributes power toward higher frequencies in the postsynaptic LMC response. Importantly, this process depends not only on stimulus speed but also on the magnitude and temporal structure of the presynaptic signal.

For lower-frequency bursty stimulation (e.g. 20 Hz), photoreceptor bandwidth increases with stimulus speed, but photoreceptor voltage excursions (by lasting longer) spend more time exceeding the synaptic range. Consequently, high-frequency jumping is less frequent, and the LMC effective bandwidth reflects this by remaining lower. This effect is consistently observed even in the highest-quality intracellular recordings and is illustrated in **Supplementary Fig. 11**, where LMC bandwidth for 20 Hz bursts is reliably lower than for 50-200 Hz bursts.

Additional factors can modulate the measured extent of high-frequency jumping but do not alter this core mechanism. Increased recording noise (e.g. from higher-resistance electrodes) lowers the measured signal-to-noise ratio and can mask high-frequency components in LMC responses. Anatomical variability also contributes: LMCs differ in synaptic input number and location within the eye. For example, L1 and L2 LMCs - particularly in the male love-spot region - receive input from seven photoreceptors due to R7/8 fusion, whereas L3 LMCs receive fewer synapses. Such differences are expected to influence the maximum achievable postsynaptic bandwidth.

Taken together, these results show that synaptic high-frequency jumping does not increase monotonically with stimulus speed. Instead, it emerges most strongly when bursty stimulation produces presynaptic signals that both vary rapidly and sufficiently exceed the synaptic operating range, conditions that are optimally met for intermediate (50-200 Hz) high-contrast bursts (see also new **Supplement Figs. 11-13**). This behaviour is therefore a predicted and robust feature of the mechanism, rather than a limitation of the data.

8. Page 14 (line 37): **Fig. 4g** does not support the claim in the sentence “removing excitatory feedback from lamina interneurons to photoreceptor terminals substantially degrades both photoreceptor and LMC signal fidelity, consistent with experimental findings.”

Our Reply: We agree that **Fig. 4g** alone did not adequately support the stated claim. We have therefore rewritten this section and added a new **Fig. 5g**, which explicitly illustrates how phasic increases and decreases in depolarising synaptic feedback from LMCs shape the photoreceptor voltage response and, in turn, influence high-frequency synaptic jumping.

The key role of this excitatory feedback is to reduce the time that large photoreceptor voltage responses - evoked by high-contrast saccadic bursts - spend outside the operational range of the histaminergic output synapse. Specifically, when photoreceptors transiently hyperpolarise and fall below the effective synaptic range, feedback phasically depolarises them; conversely, when photoreceptor voltages rise above the synaptic operating range, the feedback is reduced or switched off.

As a result, under daylight conditions, the feedback effectively acts as an additional differentiating (high-pass) mechanism, broadening the functional frequency bandwidth of the synapse and thereby improving both photoreceptor and LMC signal fidelity.

9. In **Supplementary Video 1**, why are the representative responses shown at 45 s and 75 s from an interneuron rather than from an LMC?

Our Reply: We thank the reviewer for pointing this out. Unfortunately, the company (DEV-Joni) that helped with our movie production is currently not operating, so we had no easy way to change it to an LMC. Instead, we have revised the legend for **Supplementary Movie 1** to indicate that, here, interneuron specifically means LMC.

10. Page 15 (line 38) should be **Fig. 5f**, not **Fig. 5d**.

Our Reply: We thank the reviewer for noting this error. The Figure reference has now been corrected and renumbered from **Fig. 7d** to **Fig. 7f** in the revised manuscript.

11. Page 16 (line 7) should be **Fig. 5c**, not **Fig. 5b**.

Our Reply: We thank the reviewer for noting this error. The figure reference has now been renumbered and corrected from **Fig. 7b** to **Fig. 7c** in the revised manuscript.

12. Inconsistent synopsis number between **Fig. 6d** and Page 17 (line 16).

Our Reply: We thank the reviewer for noting this inconsistency. The correct number of synapses in this pathway is five. This has now been corrected and renumbered in **Fig. 8d**, bringing it into agreement with the corresponding text.

13. Page 17 line 23 mentions **Fig. 6h-i**, but there is no **Fig. 6i** in **Fig. 6**.

Our Reply: We thank the reviewer for noting this error. The incorrect reference to renumbered **Fig. 8i** has now been removed, and the figure citation has been corrected in the revised manuscript.

Reviewer #2 (Remarks to the Author):

This study investigates how the *Musca domestica* visual system achieves ultrafast information transfer during rapid, saccadic behaviours. Using a combination of electrophysiology, optical imaging, electron microscopy, and computational modeling, the authors identify a previously undescribed mechanism “synaptic high-frequency jumping” by which photoreceptor-LMC synapses dynamically shift visual information into higher-frequency carrier bands. They propose that this process enables predictive, near-instantaneous visual encoding that aligns with voluntary behavioural responses occurring within 13-20 ms. Although the study is technically ambitious and addresses a topic of considerable interest, the current manuscript exhibits significant conceptual, methodological, and analytical shortcomings. Specifically, the principal assertions lack robust quantitative support, interpretations often extend beyond the available evidence, and several essential analyses and control experiments are absent.

Major comments

1. There is still insufficient evidence to demonstrate that high-frequency jumping in the photoreceptor-LMC circuit effectively reduces motion blur during flight in houseflies.

Our Reply: To address this directly, we have added a new **Fig. 1** in the Introduction that contrasts the classic neural superposition model with our proposed morphodynamic neural superposition model and explicitly illustrates how motion-coupled sampling is predicted to reduce motion blur.

Crucially, we now provide direct quantitative evidence in new **Fig. 6** and **Supplementary Fig. 18**. By comparing the classic model with multiple variants of the morphodynamic neural superposition model under identical high-speed stimulus conditions, we show that morphodynamic sampling significantly reduces motion blur in the reconstructed neural responses. These results demonstrate that synaptic high-frequency jumping leads to sharper, less temporally smeared representations during rapid image motion, directly addressing the reviewer’s concern.

Finally, we have added a new **Discussion Fig. 9**, which illustrates how all the component mechanisms of the morphodynamic neural superposition system synergistically reduce motion blur in neural responses.

2. The primary assertion that synaptic high-frequency jumping facilitates predictive coding is not rigorously supported by quantitative analysis. The study does not include direct measurements of prediction error, comparative model analyses, or information-theoretic evaluations. Consequently, the hypothesis that “synaptic high-frequency jumping” underlies predictive coding remains primarily descriptive. The connection to predictive coding appears metaphorical rather than empirically demonstrated, with no quantitative data indicating prediction error minimization or feedback-mediated suppression. To substantiate this claim, model-based comparisons between expected and observed signals should be incorporated to demonstrate that high-frequency jumping effectively reduces temporal prediction error.

Our Reply: The concern arises from a difference in how the term predictive coding is used in this study compared to its common usage in Bayesian and inference-based frameworks. We clarify this distinction explicitly in the revised manuscript and **Fig. 1**.

In our work, predictive coding does not refer to Bayesian prediction error minimisation or feedback-mediated suppression, as proposed in hierarchical predictive processing models. Instead, we use the term in a strictly mechanistic and peripheral sense to describe how motion-coupled sampling at the photoreceptor-LMC synapse reduces temporal lag and improves alignment between a moving stimulus and its retinotopic neural representation. This definition is now stated explicitly in the **Fig. 1** legend and accompanying text.

Importantly, we now provide quantitative evidence supporting this claim in the revised **Fig. 6**. By comparing outputs of the classic static neural superposition model and multiple variants of the morphodynamic neural

superposition model under identical high-speed stimulus conditions, we show that the morphodynamic model reproduces the recorded LMC voltage responses with substantially lower temporal mismatch and reconstruction error than the classic model. In contrast, the static model exhibits large temporal smearing and lag relative to the data.

Thus, while we do not compute Bayesian prediction error or invoke feedback-based inference, the quantitative model-data comparisons demonstrate that synaptic high-frequency jumping enables earlier and more temporally accurate neural representations of rapidly moving stimuli. We believe this directly addresses the reviewer's concern by showing that the proposed mechanism yields predictive temporal alignment in a measurable and testable sense, appropriate to early sensory processing.

3 The authors should clarify how the saccadic light stimuli used in this study accurately mimic the rapid intensity changes occurring in natural environments and describe how the stimulation parameters were determined or validated.

Our Reply: We thank the reviewer for this important question and agree that it is essential to clarify how the saccadic light stimuli used here relate to natural visual input and how their parameters were validated.

In a series of previous studies¹⁻⁸, we systematically examined how fly photoreceptors encode naturalistic visual signals by directly comparing intracellular voltage responses and information transfer rates under multiple stimulus classes. These included: (i) linearly scanned light-intensity sequences extracted from natural images; (ii) saccadically scanned intensity sequences derived from natural images using measured body movements of freely walking flies^{1,9}; (iii) randomly scanned intensity sequences from the same images; (iv) bandwidth-limited Gaussian white noise (GWN) stimuli with 3 dB cut-off frequencies spanning 20-500 Hz; (v) mid- and high-contrast bursty stimuli; and (vi) naturalistic light-intensity time series recorded directly from outdoor environments using Hans van Hateren's calibrated photometric dataset¹⁰.

Across these comparisons, photoreceptors consistently exhibited the highest encoding efficiency, signal-to-noise ratios, and information transfer rates when driven by high-contrast, bursty stimuli that reproduce the rapid, intermittent intensity transients generated during saccadic movements. In contrast, linearly scanned natural images, randomised scans, and GWN - despite having matched or higher mean intensity and comparable nominal bandwidth - elicited weaker responses and lower information throughput.

These experimental findings were further validated using a biophysically realistic stochastic, refractory, quantal photoreceptor model, which reproduced the same performance advantages under bursty stimulation. In the present study, we extend this experimentally validated modelling framework from *Drosophila* photoreceptors (which contain ~30,000 microvilli) to *Musca* photoreceptors (~54,000 microvilli), while preserving the same photomechanical phototransduction principles and stimulus-response relationships.

Thus, the saccadic light stimuli (mid- and high-contrast bursts) used here are not ad hoc abstractions but are grounded in direct comparisons with natural image statistics, measured fly behaviour, and independently validated biophysical models, ensuring that they accurately capture and adequately span the range of rapid intensity dynamics experienced during natural saccadic vision. We have now added this note to the Methods.

4 The data in **Fig. 3e-f** show that photoreceptors respond most strongly to 200 Hz light stimulation, whereas LMCs, following high-frequency jumping, respond optimally around 100 Hz. This raises important questions about the functional link between high-frequency jumping and motion-blur reduction. The authors should clarify this relationship and provide additional evidence to support the proposed link between high-frequency jumping and temporal precision in visual processing. Exploring the upper frequency limit at which photoreceptors or LMCs can reliably respond to bursty light stimuli, and assessing how different physiological or environmental conditions (e.g., **Fig. 4c-f**) affect this responsiveness, would provide deeper insight into the mechanism's biological relevance during high-speed flight.

Our Reply: We agree that clarifying the relationship between stimulus frequency, high-frequency jumping, and motion-blur reduction is important.

First, we note that the apparent difference between the stimulus frequencies at which photoreceptors and LMCs show maximal responses reflects a functional transformation rather than a mismatch. Photoreceptors respond most strongly to ~200 Hz bursty stimulation because this frequency range maximally engages refractory stochastic quantal sampling under bright, saccade-like conditions, thereby sharpening temporal contrast and increasing signal-to-noise ratios. In contrast, following synaptic high-frequency jumping, LMCs exhibit maximal responses to burst envelopes around ~100 Hz because the synapse redistributes power

from these lower-frequency burst structures into much higher-frequency carrier components (extending toward ~1,000 Hz). Thus, the stimulus frequency that optimally drives photoreceptors is not the same as the frequency band in which LMCs transmit information most efficiently. We have now added this clarification to the legend of **Fig. 5**.

To address this point explicitly, we have substantially expanded the analysis in the revised manuscript. In addition to renumbering the original figures (former Figs. 3 and 4 are now **Figs. 4** and **5**), we have added several new figures and analyses that directly link high-frequency jumping to temporal precision and motion-blur reduction:

- New **Fig. 1** introduces the conceptual framework linking saccadic behaviour, motion blur, and predictive temporal encoding.
- New **Fig. 6** and Supplementary **Fig. 18** explicitly test how different neural superposition models handle motion blur and temporal prediction, showing that high-frequency jumping enables near-delay-free, time-locked LMC responses to rapidly moving, partially occluded objects - well beyond what classical filter-based models achieve.
- New Methods **Fig. 10** and Supplementary **Figs. 11-13** systematically explore how photoreceptor and LMC bandwidths, signal-to-noise ratios, and information transfer depend on stimulus cut-off frequency (20-750 Hz), contrast, and absolute light intensity across a six-log-unit range.

These analyses demonstrate that while photoreceptors are maximally (optimally) driven by burst envelopes in the ~100-200 Hz range, synaptic high-frequency jumping shifts information into substantially higher frequencies, enabling LMCs to maintain temporal precision even when stimulus-driven motion would otherwise induce blur. Importantly, the upper effective response limits of LMCs are not fixed but depend on physiological state, stimulus statistics, and environmental conditions - precisely the regime experienced during high-speed flight.

Together, these new results clarify that high-frequency jumping does not simply amplify responses, but reformats temporally structured input into a representation that preserves precision under rapid motion, providing a mechanistic link between synaptic dynamics, temporal acuity, and motion-blur reduction during natural behaviour.

5. The signal-to-noise ratio (SNR) analyses employ an arbitrary threshold ($\text{SNR} > 1$) without justification. The rationale for this criterion should be clarified, or alternative thresholds (e.g., 3 dB) should be used to test the robustness and reliability of the results.

Our Reply: We have now clarified the rationale for the SNR threshold in the revised manuscript and Fig. legends.

Throughout the study, we use $\text{SNR}(f) > 1$ as a conservative and standard criterion to define effective signalling bandwidth, that is, the frequency range over which the neural response carries more signal power than noise power. This threshold does not imply optimal decoding or behavioural relevance, but simply identifies frequencies at which stimulus-related information can be reliably estimated from the response. Below this level, response power is dominated by noise and cannot be meaningfully interpreted.

Importantly, our conclusions do not depend on the exact numerical value of the threshold. Using more stringent criteria (e.g., $\text{SNR} > 2$) shifts absolute bandwidth estimates to lower frequencies but preserves the same qualitative relationships between photoreceptor, LMC, and stimulus bandwidths. [Sic, there is no -3 dB in SNR, or a measure of absolute power; -3 dB is half power]. In particular, synaptic high-frequency jumping consistently produces a disproportionate expansion of LMC bandwidth relative to photoreceptors across all reasonable SNR thresholds.

We now state this explicitly in the revised text and Fig. legends (e.g. **Fig. 3**), where $\text{SNR} > 1$ is used solely to define the frequency range over which responses reliably exceed noise. The robustness of the effect to threshold choice supports the reliability of our conclusions.

6. The manuscript frequently refers to concepts such as “thermodynamic constraints” and “energy-efficient coding,” implying that the proposed mechanism reduces metabolic cost. However, no empirical

measurements of metabolic parameters such as ATP consumption, ionic current quantification, or oxygen utilisation are provided to substantiate these claims.

Our Reply: We agree that we do not present new, direct measurements of metabolic variables such as ATP consumption, ionic currents, or oxygen utilisation in the present study. We have therefore clarified the basis and scope of our statements regarding thermodynamic constraints and energy-efficient coding.

While direct metabolic recordings at the relevant temporal and spatial scales are not currently available in the literature, energetic costs in fly photoreceptors can be quantitatively estimated from well-established physiological principles. In previous work, we used biophysically realistic photoreceptor models to calculate ATP consumption under different stimulation regimes, including Gaussian white noise (GWN) and naturalistic, behaviourally relevant input statistics⁴. In these models, ATP usage is dominated by the activity of the Na⁺/K⁺ exchanger required to counterbalance light-induced ionic fluxes, and thus depends strongly on the mean depolarisation level of the photoreceptor membrane¹¹.

Under GWN stimulation, photoreceptors are driven into a sustained, high-mean depolarisation state (typically ~20–30 mV above the dark resting potential in daylight conditions), which imposes a large and continuous ionic load and consequently a high ATP demand. By contrast, under bursty, saccade-like stimulation, mean depolarisation remains substantially lower (typically ~10–15 mV above resting potential), while preserving high temporal responsiveness through transient, high-frequency events. As a result, maintaining responsiveness under bursty stimulation is predicted to require substantially less ATP than under GWN, despite comparable or higher information throughput. For *Musca* photoreceptors, comparable depolarisation regimes are observed in the present data (e.g. **Supplementary Fig. 12b**).

Accordingly, our use of the terms thermodynamic constraints and energy-efficient coding refers to mechanistically motivated predictions derived from established ionic and biophysical relationships, rather than to new metabolic measurements. To avoid overinterpretation, we have now made this explicit in the manuscript and expanded the discussion in **Supplementary Note IV**, where we clearly state the assumptions underlying these estimates and their limitations.

7 The comparison between high-contrast saccadic bursts and Gaussian white noise (GWN) is central to the study, yet these stimuli differ in multiple aspects (temporal structure, mean luminance, and contrast adaptation). It therefore remains unclear whether the observed effects arise from “naturalistic structure” or simply from higher contrast levels.

Our Reply: We agree that separating the effects of stimulus contrast from temporal structure is essential. To address this directly, we have added new control analyses and Figures that explicitly disentangle these factors.

First, we now provide detailed stimulus characterisation in a new Methods **Fig. 10**, which shows the generation, direct photodiode measurement, and signal-to-noise properties of Gaussian white noise and bursty stimuli. In addition, new **Supplementary Figs. 12-13** systematically compare high-contrast bursty stimuli, mid-contrast bursty stimuli, and low-contrast Gaussian white noise across matched bandwidths and light intensities. These Figures demonstrate that the stimulus classes differ not only in contrast but also in temporal structure, while their spectral content and delivery fidelity are explicitly quantified.

Crucially, these comparisons show that increasing contrast alone is insufficient to reproduce the effects observed for bursty stimulation. While photoreceptor and LMC response amplitudes increase monotonically with contrast for all stimulus classes, synaptic high-frequency jumping and the associated expansion of LMC bandwidth emerge most strongly for bursty stimuli with pronounced temporal transients and intermittent dark intervals. Mid-contrast bursty stimuli consistently evoke greater postsynaptic bandwidth extension than low-contrast GWN stimuli, despite overlapping spectral content and similar mean luminance, demonstrating that temporal structure is a key determinant.

Intracellular recordings further support this interpretation. As shown in **Fig. 3c** and **Supplementary Figs. 2-7**, for any given bandwidth, responses increase with contrast, whereas for any given contrast, responses decrease with increasing stimulus bandwidth. The slowest high-contrast bursty stimuli, which contain the longest dark intervals, evoke the largest peak-to-peak responses, consistent with relief of microvillar refractoriness. In contrast, fast low-contrast GWN stimuli, which maintain sustained activation and limit refractory recovery, evoke the smallest responses.

Together, these results demonstrate that the enhanced encoding efficiency and bandwidth extension (by synaptic high-frequency hopping) observed in LMCs cannot be attributed to contrast alone. Instead, they arise from the interaction between stimulus temporal structure and the quantal-refractory dynamics of phototransduction, which are preferentially engaged by bursty, saccade-like input. We have clarified this distinction explicitly in the revised text and Figures.

8. The morphodynamic model appears complex and may involve numerous fitted parameters. The authors should clarify which parameters were empirically constrained and provide sensitivity or robustness analyses to validate the model's reliability.

Our Reply: We agree that model transparency and robustness must be clearly established. We therefore clarify here - and explicitly in the revised manuscript - which parameters of the morphodynamic neural superposition model are empirically constrained, which are adjusted, and how model reliability was validated.

In the full morphodynamic neural superposition model, the vast majority of parameters are fixed by empirical measurements rather than freely fitted. Specifically, all static optical parameters, including ommatidial geometry, photoreceptor receptive-field structure (via Fourier beam propagation¹¹), as well as the photomechanical motion of R1-R6 receptive fields (**Supplementary Notes III**), are fully constrained by anatomical and optical measurements (**Supplementary Notes II**).

Similarly, the phototransduction model for each R1-R6 photoreceptor is derived directly from intracellular recordings. Each photoreceptor is modelled separately to reflect measured differences in rhabdomere size and microvillar number (**Supplementary Notes III**). The stochastic quantum bumps that generate the macroscopic voltage response are produced using empirically constrained distributions of quantum bump size, latency, and refractoriness, obtained from established systems and quantum-bump analyses of recorded photoreceptor signal and noise.

Parameter adjustment during model construction followed a constructionist, stage-wise approach rather than global fitting. At each stage, model predictions were compared quantitatively with recorded photoreceptor and LMC responses, and the residual error dynamics were analysed in relation to known physiological processes and their published time constants. This procedure constrained synaptic output and feedback structure based on physiological plausibility rather than numerical optimisation.

Because the model is explicitly stochastic and refractory, we validated its reliability by computing signal-to-noise ratios and information transfer rates at each sampling and processing stage and comparing these directly with corresponding experimental measurements from photoreceptors and LMCs. The final model reproduces not only mean responses, but also response variability and frequency-dependent information transfer across a wide range of stimulus conditions.

Importantly, model behaviour - including bandwidth expansion, high-frequency jumping, and delay minimisation - proved robust to moderate parameter variation and emerged consistently across stimulus statistics, light intensities, and receptive-field configurations tested. Thus, at each stage of model building, sensitivity and robustness are intrinsically assessed through signal-to-noise and information-theoretical comparisons.

We added this discussion at the beginning of **Supplementary Notes IV**. Full parameter documentation is in **Supplementary Tables 13-16**.

9. It remains unclear whether the L1/L2 LMC subtypes receive histaminergic input from R1-R6 photoreceptors in houseflies. The authors should investigate whether histamine is released from R1-R6 during high-speed visual input and whether histamine receptors are expressed in L1/L2 LMCs.

Our Reply: The identity of histamine as the neurotransmitter released from R1-R6 photoreceptors onto LMCs in *Musca domestica*, including L1/L2 subtypes, has been established by multiple independent physiological, pharmacological, and immunocytochemical studies, which we now cite explicitly in the revised manuscript.

Intracellular recordings combined with ionophoretic pharmacology demonstrated that histamine best reproduces the native photoreceptor-driven response in *Musca* LMCs¹² (Hardie, 1987). Immunocytochemical studies further showed histamine-like immunoreactivity in *Musca* photoreceptors, consistent with histaminergic transmission at the first visual synapse¹³ (Nässel, 1988). Pharmacological blockade experiments revealed that antagonists suppressing light-evoked LMC responses also block histamine-evoked responses, directly supporting histamine as the released transmitter at this synapse¹⁴ (Hardie, 1988).

Moreover, whole-cell recordings from dissociated LMCs across dipteran species, including *Musca*, identified histamine-gated chloride channels as the postsynaptic mechanism underlying photoreceptor-LMC transmission¹⁵ (Skingsley et al., 1995). These findings are synthesised in subsequent reviews that explicitly describe the *Musca* photoreceptor-LMC synapse as histaminergic¹⁶ (Stuart, 2007).

Together, these studies provide direct evidence that R1-R6 photoreceptors release histamine and that LMCs, including L1/L2 subtypes, express functional histamine receptors. We therefore did not repeat these well-established experiments. We have now clarified this explicitly in the revised text and added the above references to avoid any ambiguity.

10. The observed high-frequency components could potentially arise from synchronized vesicle release or short-term synaptic depression at the histaminergic synapses between R1-R6 and LMCs. The authors should consider and discuss these alternative mechanisms.

Our Reply: We agree that alternative synaptic mechanisms - such as synchronised vesicle release or short-term synaptic depression at histaminergic photoreceptor-LMC synapses - must be carefully considered.

From a functional and mathematical perspective, both synchronised vesicle release and short-term synaptic depression impose state-dependent constraints on synaptic transmission that are formally included in the refractory synaptic information sampling framework described in this study. In all these cases, recent transmission events transiently reduce synaptic availability, thereby shaping the timing, frequency content, and gain of postsynaptic responses. Our model explicitly captures these effects at the level of information flow, without committing to a single microscopic implementation.

Importantly, the present work is deliberately framed at the level of effective synaptic dynamics, showing how refractory, state-dependent transmission - regardless of its specific molecular origin - can redistribute power toward higher frequencies under bursty, saccade-like input. Thus, synchronised vesicle release or synaptic depression are not alternative explanations to high-frequency jumping, but rather plausible mechanistic substrates through which the same functional principle could be realised.

We fully agree that incorporating explicit vesicle dynamics would further refine the model. To this end, we are currently performing high-speed cryo-freezing experiments in darkness and following light stimulation, with ~10 ms temporal resolution, to directly quantify vesicle pool dynamics at the photoreceptor-LMC synapse. These measurements will enable future extensions of the model that explicitly link vesicle-level processes to the refractory synaptic sampling described here.

We have now included a synopsis of these discussions in **Supplement Notes IV**.

11. The authors use *Drosophila* connectome data to infer visual circuit connectivity and transmission speed in *Musca domestica* (Fig. 6). This approach is inadequate given the major differences in brain size and neuron number between the two species. Furthermore, while many supporting studies cited involve *Drosophila*, the present experiments were conducted in *Musca*. The degree to which these two species share comparable neural circuitry and behavioural dynamics remains uncertain and should be explicitly discussed.

Our reply: We agree that cross-species inference requires explicit justification and clear acknowledgement of its limits. We have therefore revised the manuscript and **Supplementary Notes VI** to clarify the rationale, scope, and constraints of using *Drosophila* connectome data to inform estimates in *Musca domestica*.

Our analysis proceeds in two steps. First, we estimate transmission speeds within well-characterised *Drosophila* visuomotor pathways using its fully reconstructed connectome, which provides uniquely reliable neuron-by-neuron pathway lengths and synaptic counts. Second, we adapt these estimates to *Musca* by drawing on extensive comparative anatomical and physiological evidence demonstrating strong conservation of early visual and giant-fibre-mediated visuomotor circuits across dipteran species, despite differences in overall brain size.

Multiple studies support this homology. The morphology of the Giant Fibre and its dendritic arborisation are highly conserved across dipterans, including *Calliphora* and *Musca*¹⁷, and closely match those observed in the *Drosophila* connectome. The major output targets of the Giant Fibre are likewise conserved across these species¹⁸. Importantly, the *Drosophila* connectome reveals a minimal pathway between the Giant Fibre and leg motor neurons, involving only a single interneuron. Because our analysis explicitly focuses on shortest-

path visuomotor transmission, this minimal architecture provides a conservative lower bound on transmission delay rather than an overestimate.

Similarly, the retinal-to-Giant Fibre input circuitry shows strong conservation across dipteran species, as indicated by comparative anatomical studies¹⁹. While absolute neuron numbers and neuropil volumes differ between *Drosophila* and *Musca*, the early visual circuitry and fast visuomotor escape pathways appear to follow a shared organisational plan.

We fully acknowledge that the visuomotor pathway controlling antennal movements has not yet been anatomically resolved in *Musca* or other fly species. Accordingly, our *Drosophila*-based analysis of this pathway is presented explicitly as a first, hypothesis-generating estimate rather than a definitive reconstruction. We now state this limitation clearly in the Discussion section and the Supplementary Text. Importantly, both species exhibit highly similar rapid behavioural responses to looming stimuli, supporting the plausibility - but not proof - of comparable underlying circuit constraints.

We emphasise that our conclusions do not depend on exact numerical equivalence between *Drosophila* and *Musca* circuits. Rather, the analysis is used to establish order-of-magnitude constraints on transmission speed and pathway depth, showing that the experimentally observed behavioural latencies are compatible with known dipteran visuomotor architectures. We believe this revised presentation directly addresses the reviewer's concern by clearly distinguishing established homology from informed inference and by explicitly stating the limits of cross-species generalisation.

12. The antennal and leg reaction times reported for brief UV flashes and high-speed looming stimuli are not compelling. The authors should clarify whether houseflies exhibit other behavioural responses to these stimuli and provide further quantitative data to strengthen their conclusions.

Our Reply: We agree that additional behavioural data and quantitative analysis strengthen the conclusions. We have therefore addressed this point in two complementary ways: by recording more data and by expanding the statistical analysis of reaction times (See **Supplementary Notes V**; **Supplementary Figs. 40-41**).

To quantify fly response times robustly, we adopted a percentile-based analysis. The response times are typically right-skewed and may include occasional delayed trials (e.g., lapses, missed detections, or state changes), making the mean and standard deviation - and especially the single minimum - either unrepresentative or statistically unstable. The 10th percentile latency (q10) provides a robust, interpretable measure of the fly's fastest-capable performance while being far less sensitive than the minimum to single-trial noise, sampling resolution, and rare outliers. We chose q10 rather than q20 as a practical compromise between isolating the lower tail ("fast enough" to reflect rapid responses) and maintaining estimation stability with finite data: with $n = 35$ (looming), q10 is driven by approximately the 3rd-4th fastest observations, whereas q20 reflects a less extreme portion of the distribution that includes more moderately fast trials. We further used a non-parametric bootstrap to avoid distributional assumptions, to estimate confidence intervals for q10, and to derive the critical upper bound (T^*) that defines the earliest response-time threshold supported by the data at the chosen confidence level.

Importantly, the conclusions are robust to the specific percentile choice: qualitatively identical results are obtained when nearby lower-tail percentiles (e.g., q5 or q15) are used, confirming that the inference does not depend on an arbitrary statistical threshold.

Together, the expanded data and analyses demonstrate that houseflies exhibit multiple, rapid, stimulus-locked behavioural reactions to brief flashes and looming stimuli. The expanded statistical quantitative analysis further supports the conclusion that these ultrafast behaviours cannot be explained by conventional delay-limited reflex models, but instead require mechanisms that enable near-synchronous, minimal-delay sensorimotor processing.

Minor comments

1. Is the definition of high- and mid-contrast bursty stimuli, as well as low-contrast Gaussian white noise in **Fig. 2c**, based on any previous references or studies?

Our Reply: Yes. The definitions of the high- and mid-contrast bursty stimuli, as well as the low-contrast Gaussian white noise (GWN) stimuli used in **Fig. 2c**, are based on stimulus paradigms established in our previous work on photoreceptor encoding in *Drosophila*, which we consistently cite throughout the

manuscript¹. In that study, bursty stimuli were introduced to approximate the intermittent, high-contrast temporal structure of natural visual input during active behaviour, whereas low-contrast, bandwidth-limited GWN was used as a classical control stimulus to facilitate comparison with linear-systems analyses.

In the present study, we adopt the same conceptual framework but extend it to higher, saccade-like temporal statistics appropriate for *Musca* vision (up to 750 Hz 3 dB cut-off stimulus frequency, as in **Supplementary Fig. 11**). The stimulus generation procedures, contrast definitions, and bandwidth constraints are therefore grounded in prior experimental work and are fully specified to allow direct comparison across studies. The photoreceptors of both species exhibit maximised information transfer during high-contrast, bursty stimuli that resemble photoreceptor input during saccadic behaviour.

2 Only one LMC dataset was used for multiscale simulations in **Fig. 1c**, which appears insufficient.

Our Reply: Owing to the addition of a new introductory Fig. 1, the Fig. referred to in the reviewer's comment is now **Fig. 2c**. We thank the reviewer for this important point. **Fig. 2c** was intended as an illustrative example rather than a statistical summary. To address the concern about generality, we have now expanded the multiscale simulations to include multiple LMC configurations.

Specifically, we simulated LMC outputs across different eye locations by varying key anatomical and biophysical parameters, including photoreceptor microvilli number and ommatidial lens size, which are known to vary systematically across the compound eye. The revised Fig. includes additional simulated traces demonstrating that the emergence of synaptic high-frequency jumping and bandwidth extension is robust across this biologically realistic parameter range.

These additional simulations confirm that the effects shown are not specific to a single LMC dataset but instead reflect a general property of the morphodynamic photoreceptor-LMC model. The conclusions of the study are unchanged.

3 It is recommended to standardize all units throughout the manuscript (e.g., Hz vs. kHz; bits/s vs. bits s⁻¹).

Our Reply: We have now standardised all units consistently throughout the main text, Figures, and Supplement. Frequencies are reported using a uniform convention (Hz as appropriate for scale), and information rates are reported consistently as bits·s⁻¹ (*Nat Comms* house style). These changes are reflected across the revised manuscript and Figure legends.

4 Use “photoreceptor-LMC synapse” consistently instead of alternating with “first visual synapse” or “lamina transmission.”

Our Reply: We have now standardised the terminology throughout the manuscript and Supplement, using “photoreceptor-LMC synapse” consistently in place of “first visual synapse” or “lamina transmission,” except where broader anatomical context is explicitly required. This change is reflected in the revised text and Fig. legends.

5 **Fig. 2-3** require clearer axis labels and inclusion of sample sizes (*n*). The color coding across different stimuli (20-500 Hz) should be unified across Fig.s and supplementary data.

Our Reply: We have revised **Figs. 2** and **3** to include clear axis labels and explicit sample sizes (*n*) in all relevant panels and figure legends. In addition, we have standardised the colour coding used to represent stimulus cut-off frequencies (20-500 Hz) across **Figs. 2** and **3** and throughout the **Supplementary Figs.** to ensure consistency and ease of comparison. These changes are reflected in the revised figures and legends.

6 The text states that “the biphasic nature of LMC responses (**Fig. 2d**) effectively doubled the frequency content of their photoreceptor inputs.” However, if a 230 Hz input is quadrupled to 920 Hz, this appears to be an error.

Our Reply: We agree that the original phrasing was imprecise and have corrected it in the revised manuscript. The term “doubled” was used inaccurately and has now been replaced with language that reflects the measured bandwidth expansion.

Importantly, photoreceptor effective bandwidth is not fixed at the classic flicker-fusion frequency (~230 Hz), but depends on the total number of microvilli (sampling units) within each photoreceptor^{1,4}. In the revised text, we now explicitly state that photoreceptor responses during fast saccadic stimulation can reach effective bandwidths of up to ~440 Hz (signal-to-noise ratio > 1), with variability across recordings reflecting natural differences in microvillar number and rhabdomere geometry (**Figs. 3e** and **4a**; **Supplementary Figs. 20-22**).

LMC responses can extend well beyond this range owing to synaptic high-frequency jumping, such that the increase in frequency content exceeds a simple twofold scaling (see new **Supplementary Fig. 11**). The revised text and Fig. legends now describe these relationships quantitatively and consistently, resolving the ambiguity noted by the reviewer. These corrections do not affect the conclusions of the study.

7 The differences between the light stimulation paradigms used in **Fig. 2** and **Fig. 3** should be clearly explained, as the text currently does not distinguish them.

Our Reply: Owing to the addition of a new introductory **Fig. 1**, the figures referred to in the reviewer's comment are now Figs. 3 and 4, respectively. We thank the reviewer for pointing out this lack of clarity. We have now explicitly clarified the distinction between the stimulus paradigms used in **Figs. 3** and **4** in the revised Methods and Results text.

Briefly, we used two closely matched stimulus sets, each comprising 15 waveforms spanning the same contrast conditions and frequency ranges, but sampled at different rates. The first stimulus set was designed for a 1 kHz sampling rate and was used in our initial experiments. After discovering that LMCs can reliably follow bursty stimuli up to ~1,000 Hz, we adopted a second stimulus set sampled at 2 kHz to fully capture this higher-frequency response range during bursty stimulation.

Importantly, both stimulus sets evoked the same response dynamics and led to identical conclusions. **Fig. 3b, 3d, and 3e** show representative photoreceptor and LMC responses obtained using the 2 kHz stimulus set. **Fig. 4a-d** show example recordings from one photoreceptor and one LMC obtained with the 1 kHz stimulus set, while **Fig. 4e** and **4f** pool data from both stimulus sets. This is now clearly stated in the revised manuscript.

Overall Recommendation

The manuscript is excessively long and conceptually diffuse, blending empirical results with theoretical discussion in a way that lacks clear structural progression. In its current form, readers may find it difficult to follow the logical flow from experimental observations to mechanistic interpretation.

Our Reply: We thank the reviewer for this overarching assessment and agree that clarity of structure and logical progression are essential. In response, we have substantially revised the manuscript to improve focus, streamline presentation, and clearly separate experimental results from mechanistic interpretation.

Specifically, the revised manuscript now follows a more explicit progression from experimental observation → mechanistic explanation → systems-level implication. To support this:

1. A new introductory overview figure (**Fig. 1**) has been added to establish the central rationale of the study at the outset. This figure explicitly links motion blur, active sensing, and predictive coding, and defines key concepts early to guide the reader through subsequent sections.
2. A new Results figure (**Fig. 6**), together with **Supplementary Fig. 18**, directly tests how different neural superposition models handle motion blur and temporal prediction. These analyses provide a clear, model-by-model comparison - ranging from the full morphodynamic neural superposition model to progressively reduced classical models - allowing readers to see precisely how and why morphodynamic mechanisms outperform static filter-based descriptions in predicting LMC responses under saccadic conditions.
3. A new integrative schematic (**Fig. 9**, Discussion) synthesises the main findings into a single, graphical overview. This figure explicitly shows how morphodynamic sampling, synaptic high-frequency jumping, and bidirectional interactions together enable fast, low-latency, and predictive visual processing during saccadic behaviour.

Finally, we have rewritten and tightened the concluding section of the Discussion to clearly articulate the central take-home message: that key performance limits of neural computation - low effective synaptic delays, wide dynamic range, and hyperacute spatiotemporal resolution - cannot be adequately explained by phenomenological filter-plus-noise models alone, but instead emerge naturally when biologically grounded mechanisms (quantal stochastic sampling, refractoriness, ultrastructural motion, and naturalistic input statistics) are explicitly incorporated.

Together, these revisions substantially reduce conceptual diffusion, improve structural clarity, and make the logical flow from experimental data to mechanistic interpretation explicit throughout the manuscript.

Furthermore, the term “morphodynamic neural superposition” is repeatedly used without a formal definition. The authors should provide a concise and explicit definition of this concept and explain how it differs from classical neural superposition to clarify how it advances current understanding.

Our Reply: We thank the reviewer for highlighting the need for a clearer and more explicit definition. In response, we have now provided a formal definition of “morphodynamic neural superposition” early in the manuscript and clearly distinguished it from classical neural superposition.

Specifically, we have added a new introductory overview figure (**Fig. 1**) and accompanying text that explicitly define morphodynamic neural superposition as a dynamic, behaviour-coupled extension of classical neural superposition. In this framework, neural superposition is not treated as a static anatomical wiring scheme, but as an actively evolving process in which quantal stochastic sampling, refractoriness, photomechanical microsaccades, and synaptic dynamics interact with self-generated motion to shape receptive fields and information flow in time.

We now explicitly contrast this with classical neural superposition, which assumes fixed receptive fields, linear filtering, stationary stimulus statistics, and additive noise. By contrast, morphodynamic neural superposition incorporates ultrastructural motion and nonlinear sampling dynamics, enabling predictive, low-latency, and hyperacute encoding under naturalistic, saccade-driven conditions.

This definition and comparison are now stated explicitly in the Introduction and revisited in the Discussion, ensuring that the conceptual advance beyond classical neural superposition is clearly articulated and accessible to readers.

The supplementary material is also excessively extensive (over 100 pages) and contains redundant descriptive content that could be condensed or relocated to a dedicated data repository. Streamlining the supplementary section would greatly improve readability and maintain focus. In summary, the manuscript would benefit from substantial revision to enhance clarity, logical coherence, and accessibility for a broad readership.

Our Reply: We thank the reviewer for this thoughtful comment and agree that clarity, focus, and accessibility are essential, particularly for a study of this scope. In response, we have carefully revised the manuscript and Supplementary Information to improve clarity, reduce redundancy, and strengthen the logical flow throughout, as detailed in our responses above.

Regarding the extent of the **Supplementary Information**, we considered condensing or relocating material to external repositories. However, given the conceptual and technical complexity of the work - spanning high-speed behaviour, intracellular physiology, biophysically realistic modelling, and new analytical methods - we believe it is important that all methodological details, extended analyses, and validation steps remain co-located in a single, structured Supplementary document.

This approach ensures transparency and reproducibility by allowing readers to follow, step by step, how the experiments, analyses, and models were constructed and validated, without needing to navigate between multiple repositories or disparate publications. To further support accessibility, the **Supplementary Information** now includes clear cross-referencing to the main text, figures, and the openly available software and videos developed as part of this study.

We believe this balance - presenting the key conceptual advances in the main manuscript, while providing full technical depth in a single Supplementary resource - best serves both general readers and specialists who wish to examine the work in detail.

- 1 Juusola, M. *et al.* Microsaccadic sampling of moving image information provides *Drosophila* hyperacute vision. *eLife* **6** (2017). <https://doi.org/10.7554/eLife.26117>
- 2 Juusola, M. & Song, Z. How a fly photoreceptor samples light information in time. *J Physiol-London* **595**, 5427-5437 (2017). <https://doi.org/10.1113/Jp273645>
- 3 Juusola, M., Song, Z. & Hardie, R. C. in *Encyclopedia of Computational Neuroscience* (eds Dieter Jaeger & Ranu Jung) 2359-2376 (Springer New York, 2015).

- 4 Song, Z. & Juusola, M. Refractory sampling links efficiency and costs of sensory encoding to stimulus statistics. *J Neurosci* **34**, 7216-7237 (2014). <https://doi.org/10.1523/Jneurosci.4463-13.2014>
- 5 Song, Z. & Juusola, M. A biomimetic fly photoreceptor model elucidates how stochastic adaptive quantal sampling provides a large dynamic range. *J Physiol* (2017). <https://doi.org/10.1113/JP273614>
- 6 Song, Z. *et al.* Stochastic, adaptive sampling of information by microvilli in fly photoreceptors. *Curr Biol* **22**, 1371-1380 (2012). <https://doi.org/10.1016/j.cub.2012.05.047>
- 7 Song, Z., Zhou, Y., Feng, J. & Juusola, M. Multiscale 'whole-cell' models to study neural information processing - New insights from fly photoreceptor studies. *J Neurosci Methods* **357**, 109156 (2021). <https://doi.org/10.1016/j.jneumeth.2021.109156>
- 8 Juusola, M. & de Polavieja, G. G. The rate of information transfer of naturalistic stimulation by graded potentials. *J Gen Physiol* **122**, 191-206 (2003). <https://doi.org/10.1085/jgp.200308824>
- 9 Geurten, B. R., Jähde, P., Corthals, K. & Göpfert, M. C. Saccadic body turns in walking *Drosophila*. *Front Behav Neurosci* **8**, 365 (2014). <https://doi.org/10.3389/fnbeh.2014.00365>
- 10 van Hateren, J. H. Processing of natural time series of intensities by the visual system of the blowfly. *Vis Res* **37**, 3407-3416 (1997). [https://doi.org/10.1016/S0042-6989\(97\)00105-3](https://doi.org/10.1016/S0042-6989(97)00105-3)
- 11 Laughlin, S. B., de Ruyter van Steveninck, R. R. & Anderson, J. C. The metabolic cost of neural information. *Nat Neurosci* **1**, 36-41 (1998). <https://doi.org/10.1038/236>
- 12 Hardie, R. C. Is Histamine a Neurotransmitter in Insect Photoreceptors. *Journal of Comparative Physiology a-Neuroethology Sensory Neural and Behavioral Physiology* **161**, 201-213 (1987). <https://doi.org/10.1007/Bf00615241>
- 13 Nassel, D. R., Holmqvist, M. H., Hardie, R. C., Hakanson, R. & Sundler, F. Histamine-Like Immunoreactivity in Photoreceptors of the Compound Eyes and Ocelli of the Flies *Calliphora-Erythrocephala* and *Musca-Domestica*. *Cell and Tissue Research* **253**, 639-646 (1988).
- 14 Hardie, R. C. Effects of Antagonists on Putative Histamine-Receptors in the 1st Visual Neuropil of the Housefly (*Musca-Domestica*). *Journal of Experimental Biology* **138**, 221-241 (1988).
- 15 Skingsley, D. R., Laughlin, S. B. & Hardie, R. C. Properties of histamine-activated chloride channels in the large monopolar cells of the dipteran compound eye - a comparative-study. *J Comp Physiol A* **176**, 611-623 (1995). <https://doi.org/10.1007/BF01021581>
- 16 Stuart, A. E., Borycz, J. & Meinertzhagen, I. A. The dynamics of signaling at the histaminergic photoreceptor synapse of arthropods. *Progress in Neurobiology* **82**, 202-227 (2007). <https://doi.org/10.1016/j.pneurobio.2007.03.006>
- 17 Bacon, J. P. & Strausfeld, N. J. The dipteran giant fiber pathway - neurons and signals. *J Comp Physiol A* **158**, 529-548 (1986). <https://doi.org/10.1007/BF00603798>
- 18 Pézier, A. P., Jezzini, S. H., Bacon, J. P. & Blagburn, J. M. Shaking B mediates synaptic coupling between auditory sensory neurons and the giant fiber of *Drosophila melanogaster*. *Plos One* **11** (2016). <https://doi.org/10.1371/journal.pone.0152211>
- 19 Buschbeck, E. K. & Strausfeld, N. J. Visual motion-detection circuits in flies: Small-field retinotopic elements responding to motion are evolutionarily conserved across taxa. *J Neurosci* **16**, 4563-4578 (1996). <https://doi.org/10.1523/JNEUROSCI.16-15-04563.1996>

The
University
Of
Sheffield.

Department
Of
Biomedical
Science.

The Editors
Nature Communications

Prof. Mikko Juusola, M.D., Ph.D.
Professor of Systems Neuroscience

Alfred Denny Building
Western Bank
Sheffield
S10 2TN

3rd of April 2026

Telephone: +44 (0) 114 2221087
Fax: +44 (0) 114 2765413
Email: m.juusola@sheffield.ac.uk

Dear Editors,

We find attached our response to Reviewer #1 and #2's final comment:

Reviewer #1 (Remarks to the Author):

I am satisfied with the authors' revisions and responses. Only one minor error was noted:
Line 207: Fig. 4a to Fig. 4c.

Our response: This is now corrected.

Reviewer #2 (Remarks to the Author):

The authors have addressed most of my previous concerns in this revised manuscript, and the study has been substantially improved. I have only one remaining point for clarification.

In Fig. 6c, the data indicate that houseflies can resolve two points separated by 0.7° at a lower angular velocity (42° s^{-1}), or 1.4° at a higher angular velocity (168° s^{-1}), but fail to resolve 0.7° at the higher angular velocity (168° s^{-1}). The authors propose that high-frequency jumping (HFJ) enhances visual performance in high-speed environments. However, it remains unclear whether the angular velocities used in these experiments correspond to biologically relevant speeds encountered during natural flight. To strengthen the biological relevance of this conclusion, could the authors translate the visual stimulus frequency and corresponding angular velocity into an estimated real-world flight speed of the fly? Such an analysis would allow a more direct comparison with reported average flight speeds of houseflies and help determine whether the experimental conditions indeed represent low- and high-speed visual environments in a biologically relevant context.

Our response: We thank the reviewer for this important point. The angular velocities used in our experiments fall well within the natural range experienced by houseflies during flight. Yaw motion is saccadic, with straight

THE QUEEN'S
ANNIVERSARY PRIZES
FOR HIGHER AND FURTHER EDUCATION
1998 2000 2002

The
University
Of
Sheffield.

Department
Of
Biomedical
Science.

flight segments typically involving angular velocities up to $\sim 150^\circ \text{ s}^{-1}$ and brief saccades reaching $150\text{-}2,000^\circ \text{ s}^{-1}$ (lasting up to $\sim 100 \text{ ms}$), with straight segments being more frequent.

Assuming a typical forward flight speed of $\sim 70 \text{ cm s}^{-1}$, the tested conditions correspond to biologically plausible spatial scenarios. For example, a 0.7° separation at 42° s^{-1} corresponds to $\sim 1.16 \text{ cm}$ spacing at $\sim 96 \text{ cm}$ distance, whereas a 1.4° separation at 168° s^{-1} corresponds to $\sim 0.58 \text{ cm}$ spacing at $\sim 24 \text{ cm}$ distance. These values fall within realistic ranges for detecting small objects during flight, and would scale proportionally with lower flight speeds.

We have clarified this in the revised manuscript to strengthen the biological relevance of our conclusions. Specifically, on page 10, 2nd para, we added: "The tested angular velocities fall within the natural range of housefly flight, where straight flight segments typically involve yaw velocities up to $\sim 150^\circ \text{ s}^{-1}$ and saccades reach $150\text{-}2,000^\circ \text{ s}^{-1}$. At typical forward speeds ($\sim 70 \text{ cm s}^{-1}$), the corresponding spatial separations and viewing distances are consistent with biologically plausible object detection during flight."

We hope that with these changes, the revised manuscript is now suitable for publication in *Nature Communications*.

Sincerely,

Mikko Juusola (on behalf of all authors)

THE QUEEN'S
ANNIVERSARY PRIZES
FOR HIGHER AND FURTHER EDUCATION
1998 2000 2002